# AMORTIZED NESTEROV'S MOMENTUM: ROBUST AND LIGHTWEIGHT MOMENTUM FOR DEEP LEARNING

## ABSTRACT

Stochastic Gradient Descent (SGD) with Nesterov's momentum is a widely used optimizer in deep learning, which is observed to have excellent generalization performance. In this work, we propose *Amortized Nesterov's Momentum*, which is a special variant of Nesterov's momentum. Compared with Nesterov's momentum, our new momentum has more robust iterates and higher efficiency. Our empirical results show that it achieves faster early convergence and comparable final generalization performance with little-to-no tuning. Just like Nesterov's method, the new schemes are also proved optimal in general convex setting. Our analysis sheds light on the understanding of the new variant.

## 1 INTRODUCTION

In recent years, Gradient Descent (GD) (Cauchy, 1847) and its variants have been widely used to solve large scale machine learning problems. Among them, Stochastic Gradient Descent (SGD) (Robbins & Monro, 1951), which replaces gradient with an unbiased stochastic gradient estimator, is a popular choice of optimizer especially for neural network training which requires lower precision. Sutskever et al. (2013) found that using SGD with Nesterov's momentum (Nesterov, 1983; 2013b), which was originally designed to accelerate deterministic convex optimization, achieves substantial speedups for training neural networks. This finding essentially turns SGD with Nesterov's momentum into the benchmarking method of neural network design, especially for classification tasks (He et al., 2016b;a; Zagoruyko & Komodakis, 2016; Huang et al., 2017). It is observed that in these tasks, the momentum technique plays a key role in achieving good generalization performance.

Adaptive methods (Duchi et al., 2011; Kingma & Ba, 2015; Tieleman & Hinton, 2012; Reddi et al., 2018), which are also becoming increasingly popular in the deep learning community, diagonally scale the gradient to speed up training. However, Wilson et al. (2017) show that these methods always generalize poorly compared with SGD with momentum (both classical momentum (Polyak, 1964) and Nesterov's momentum).

In this work, we introduce *Amortized Nesterov's Momentum*, which is a special variant of Nesterov's momentum. From users' perspective, the new momentum has only one additional integer hyper-parameter $m$ to choose, which we call the amortization length. Learning rate and momentum parameter of this variant are strictly aligned with Nesterov's momentum and by choosing $m = 1$, it recovers Nesterov's momentum. This paper conducts an extensive study based on both empirical evaluation and convex analysis to identify the benefits of the new variant (or from users' angle, to set $m$ apart from 1). We list the advantages of Amortized Nesterov's Momentum as follows:

- Increasing $m$ improves robustness[1]. This is an interesting property since the new momentum not only provides acceleration, but also enhances the robustness. We provide an understanding of this property by analyzing the relation between convergence rate and $m$ in the convex setting.

- Increasing $m$ reduces (amortized) iteration complexity.

- A suitably chosen $m$ boosts the convergence rate in the early stage of training and produces comparable final generalization performance.

---

[1]In this work, *robustness* refers to the probability of an optimizer significantly deviating from its expected performance, which can be reflected by the deviations of accuracy or loss in the training process over multiple runs that start with the same initial guess.

- It is easy to tune $m$. The performances of the methods are stable for a wide range of $m$ and we prove that the methods converge for any valid choice of $m$ in the convex setting.

- If $m$ is not too large, the methods obtain the optimal convergence rate in general convex setting, just like Nesterov's method.

The new variant does have some minor drawbacks: it requires one more memory buffer, which is acceptable in most cases, and it shows some undesired behaviors when working with learning rate schedulers, which can be addressed by a small modification. Considering these pros and cons, we believe that the proposed variant can benefit many large-scale deep learning tasks.

Our high level idea is simple: the stochastic Nesterov's momentum can be unreliable since it is provided only by the previous stochastic iterate. The iterate potentially has large variance, which may lead to a false momentum that perturbs the training process. We thus propose to use the stochastic Nesterov's momentum based on several past iterates, which provides robust acceleration. In other words, instead of immediately using an iterate to provide momentum, we put the iterate into an "amortization plan" and use it later.

## 2 PRELIMINARIES: SGD AND NESTEROV'S MOMENTUM

We start with a review of SGD and Nesterov's momentum. We discuss some subtleties in the implementation and evaluation, which contributes to the interpretation of our methods.

**Notations**  In this paper, we use $x \in \mathbb{R}^d$ to denote the vector of model parameters. $\|\cdot\|$ and $\langle \cdot, \cdot \rangle$ denote the standard Euclidean norm and inner product, respectively. Scalar multiplication for $v \in \mathbb{R}^d$ and $\beta \in \mathbb{R}$ is denoted as $\beta \cdot v$. $f : \mathbb{R}^d \to \mathbb{R}$ denotes the loss function to be minimized and $\nabla f(x)$ represents the gradient of $f$ evaluated at $x$. We denote the unbiased stochastic gradient estimator of $\nabla f(x)$ as $\nabla f_i(x)$ with the random variable $i$ independent of $x$ (e.g., using mini-batch). We use $x_0 \in \mathbb{R}^d$ to denote the initial guess.

**SGD**  SGD has the following simple iterative scheme, where $\gamma \in \mathbb{R}$ denotes the learning rate:

$$x_{k+1} = x_k - \gamma \cdot \nabla f_{i_k}(x_k), \text{ for } k \geq 0.$$

**Nesterov's momentum**  The original Nesterov's accelerated gradient (with constant step) (Nesterov, 1983; 2013b) has the following scheme[2] ($y \in \mathbb{R}^d, \eta, \beta \in \mathbb{R}$ and $y_0 = x_0$):

$$
\begin{aligned}
y_{k+1} &= x_k - \eta \cdot \nabla f(x_k), \\
x_{k+1} &= y_{k+1} + \beta \cdot (y_{k+1} - y_k), \text{ for } k \geq 0,
\end{aligned}
\tag{1}
$$

where we call $\beta \cdot (y_{k+1} - y_k)$ the momentum. By simply replacing $\nabla f(x_k)$ with $\nabla f_{i_k}(x_k)$, we obtain the SGD with Nesterov's momentum, which is widely used in deep learning. To make this point clear, recall that the reformulation in Sutskever et al. (2013) (scheme (2), also the Tensorflow (Abadi et al., 2016) version) and the PyTorch (Paszke et al., 2017) version (scheme (3)) have the following schemes ($v, v^{pt} \in \mathbb{R}^d$ and $v_0 = v_0^{pt} = \mathbf{0}$): for $k \geq 0$,

$$
(2) \begin{cases} v_{k+1} = \beta \cdot v_k - \eta \cdot \nabla f_{i_k}(y_k + \beta \cdot v_k), \\ y_{k+1} = y_k + v_{k+1}. \end{cases} \qquad (3) \begin{cases} v_{k+1}^{pt} = \beta \cdot v_k^{pt} + \nabla f_{i_k}(x_k), \\ x_{k+1} = x_k - \eta \cdot (\beta \cdot v_{k+1}^{pt} + \nabla f_{i_k}(x_k)). \end{cases}
$$

Here the notations are modified based on their equivalence to scheme (1). It can be verified that schemes (2) and (3) are equivalent to (1) through $v_k = \beta^{-1} \cdot (x_k - y_k)$ and $v_k^{pt} = \eta^{-1} \beta^{-1} \cdot (y_k - x_k)$, respectively (see Defazio (2018) for other equivalent forms of scheme (1)).

Interestingly, both PyTorch and Tensorflow[3] track the values $\{x_k\}$, which we refer to as **M-SGD**. This choice allows a consistent implementation when wrapped in a generic optimization layer (Defazio, 2018). However, the accelerated convergence rate (in the convex case) is built upon $\{y_k\}$ (Nesterov, 2013b) and $\{x_k\}$ may not possess such a theoretical improvement. We use **OM-SGD** to refer to the Original M-SGD that outputs $\{y_k\}$.

---

[2] We exchange the notations of $x$ and $y$ in Nesterov (2013b).

[3] Tensorflow tracks the values $\{y_k + \beta \cdot v_k\} = \{x_k\}$.

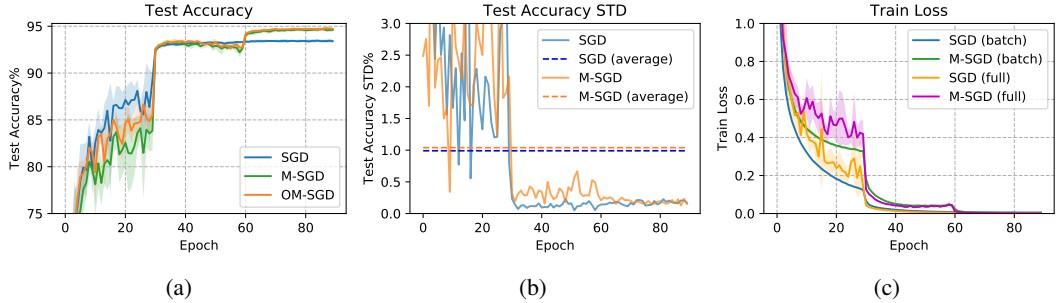

Figure 1: ResNet34 on CIFAR-10. Initial learning rate $\eta_0 = 0.1$, momentum $\beta = 0.9$, run 5 seeds (same $x_0$). In (a) (c), we plot mean curves with shaded bands indicating $\pm 1$ standard deviation. (b) shows the standard deviation of test accuracy and its average over 90 epochs. Best viewed in color.

**SGD and M-SGD**    In order to study the features of momentum, in this work, we regard momentum as an add-on to plain SGD, which corresponds to fixing the learning rates[4] $\gamma = \eta$. From the interpretation in Allen-Zhu & Orecchia (2017), $\eta$ represents the learning rate for the gradient descent "inside" Nesterov's method. To introduce the evaluation metrics of this paper, we report the results of training ResNet34 (He et al., 2016b) on CIFAR-10 (Krizhevsky et al., 2009) (our basic case study) using SGD and M-SGD in Figure 1. In this paper, all the multiple runs start with the same initial guess $x_0$. Figure 1a shows that Nesterov's momentum hurts the convergence in the first 60 epochs but accelerates the final convergence, which verifies the importance of momentum for achieving high accuracy. Figure 1b depicts the robustness of M-SGD and SGD, which suggests that adding Nesterov's momentum slightly increases the uncertainty in the training process of SGD.

**Train-batch loss vs. Full-batch loss**    In Figure 1c, train-batch loss stands for the average of batch losses forwarded in an epoch, which is commonly used to indicate the training process in deep learning. Full-batch loss is the average loss over the entire training dataset evaluated at the end of each epoch. In terms of optimizer evaluation, full-batch loss is much more informative than train-batch loss as it reveals the robustness of an optimizer. However, full-batch loss is too expensive to evaluate and thus we only measure it on small datasets. On the other hand, test accuracy couples optimization and generalization, but since it is also evaluated at the end of the epoch, its convergence is similar to full-batch loss. Considering the basic usage of momentum in deep learning, we mainly use test accuracy to evaluate optimizers. We provide more discussion on this issue in Appendix C.2.

**M-SGD vs. OM-SGD**    We also include OM-SGD in Figure 1a. In comparison, the final accuracies of M-SGD and OM-SGD are $94.606\% \pm 0.152\%$ and $94.728\% \pm 0.111\%$ with average deviations at $1.040\%$ and $0.634\%$, respectively. This difference can be explained following the interpretation in Hinton (2012) that $\{x_k\}$ are the points after "jump" and $\{y_k\}$ are the points after "correction".

## 3    AMORTIZED NESTEROV'S MOMENTUM

In this section, we formally introduce SGD with Amortized Nesterov's Momentum (AM1-SGD) in Algorithm 1 with the following remarks:

**Options**    It can be verified that if $m = 1$, AM1-SGD with Option I degenerates to M-SGD and Option II corresponds to OM-SGD. Just like the case for M-SGD and OM-SGD, the accelerated convergence rate is built upon Option II while Option I is easier to be implemented in a generic optimization layer[5]. Intuitively, Option I is SGD with amortized momentum and Option II applies an $m$-iterations tail averaging on Option I.

---

[4]Ma & Yarats (2019) observed that when effective learning rates $\gamma = \eta(1 - \beta)^{-1}$ are fixed, M-SGD and SGD have similar performance. We provide a discussion on this observation in Appendix C.1.

[5]To implement Option II, we can either maintain another identical network for the shifted point $\tilde{x}$ or temporarily change the network parameters in the evaluation phase.

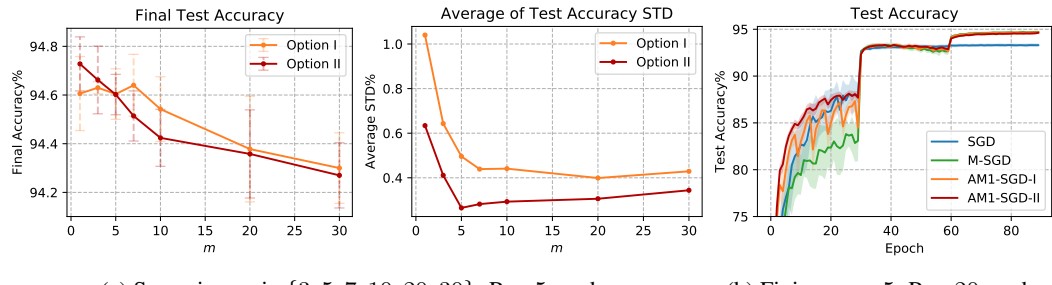

(a) Sweeping $m$ in $\{3, 5, 7, 10, 20, 30\}$. Run 5 seeds.  (b) Fixing $m = 5$. Run 20 seeds.

Figure 2: ResNet34 on CIFAR-10. For all methods, $\eta_0 = 0.1, \beta = 0.9$. Labels of AM1-SGD are 'AM1-SGD-$\{Option\}$'. Shaded bands (or bars) indicate $\pm 1$ standard deviation. Best viewed in color.

---

**Algorithm 1** AM1-SGD

**Input:** Initial guess $x_0$, learning rate $\eta$, momentum $\beta$, amortization length $m$, iteration number $K$.
**Initialize:** $x \leftarrow x_0, \tilde{x} \leftarrow x_0, \tilde{x}^+ \leftarrow \mathbf{0}$ {a running average}.
 1: **for** $k = 0, \dots, K - 1$ **do**
 2:     $x \leftarrow x - \eta \cdot \nabla f_{i_k}(x)$.
 3:     $\tilde{x}^+ \leftarrow \tilde{x}^+ + \frac{1}{m} \cdot x$.
 4:     **if** $(k + 1) \mod m = 0$ **then**
 5:        $x \leftarrow x + \beta \cdot (\tilde{x}^+ - \tilde{x})$. {adding amortized momentum}
 6:        $\tilde{x} \leftarrow \tilde{x}^+, \tilde{x}^+ \leftarrow \mathbf{0}$.
 7:     **end if**
 8: **end for**
**Output:** Option I: $x$, Option II: $\tilde{x}$.        * The symbol '$\leftarrow$' denotes assignment.

---

**Efficiency** We can improve the efficiency of Algorithm 1 by maintaining a running scaled momentum $\tilde{v}^+ \triangleq m \cdot (\tilde{x}^+ - \tilde{x})$ instead of the running average $\tilde{x}^+$, by replacing the following steps in Algorithm 1:

**Initialize:**    $x \leftarrow x_0, \tilde{x} \leftarrow x_0, \tilde{v}^+ \leftarrow -m \cdot x_0,$    **Step 3:**    $\tilde{v}^+ \leftarrow \tilde{v}^+ + x$.

**Step 5:**    $x \leftarrow x + (\beta/m) \cdot \tilde{v}^+$.            **Step 6:**    $\tilde{x} \leftarrow \tilde{x} + (1/m) \cdot \tilde{v}^+, \tilde{v}^+ \leftarrow -m \cdot \tilde{x}$.

Then, in one $m$-iterations loop, for each of the first $m - 1$ iterations, AM1-SGD requires 1 vector addition and 1 scaled vector addition. At the $m$-th iteration, it requires 1 vector addition, 1 scalar-vector multiplication and 3 scaled vector additions. In comparison, M-SGD (standard PyTorch) requires 1 vector addition, 1 (in-place) scalar-vector multiplication and 2 scaled vector additions per iteration. Thus, as long as $m > 2$, AM1-SGD has lower amortized cost than M-SGD. For memory complexity, AM1-SGD requires one more auxiliary buffer than M-SGD.

**Tuning $m$** We did a parameter sweep for $m$ in our basic case study. We plot the final and the average deviation of test accuracies over 5 runs against $m$ in Figure 2a. Note that $m = 1$ corresponds to the results of M-SGD and OM-SGD, which are already given in Figure 1. From this empirical result, $m$ introduces a trade-off between final accuracy and robustness (the convergence behaviors can be found in Appendix A.1). Figure 2a suggests that $m = 5$ is a good choice for this task. For simplicity, and also as a recommended setting, we fix $m = 5$ for the rest of experiments in this paper.

**A momentum that increases robustness** To provide a stronger justification, we ran 20 seeds with $m = 5$ in Figure 2b and the detailed data are given in Figure 3 & Table 1. The results show that the amortized momentum significantly increases the robustness. Intuitively, the gap between Option I and Option II can be understood as the effect of tail averaging. However, the large gap between Option I and SGD is somewhat mysterious: what Option I does is to inject a very large momentum[6] into SGD every $m$ iterations. It turns out that this momentum not only provides acceleration, but also helps the algorithm become more robust than SGD. This observation basically differentiates AM1-SGD from a simple interpolation in-between M-SGD and SGD.

---

[6]Amortized momentum $\beta \cdot (\tilde{x}^+ - \tilde{x})$ is expected to be much large than Nesterov's momentum $\beta \cdot (y_{k+1} - y_k)$.

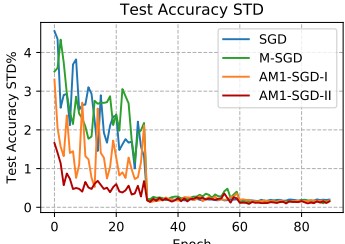

| METHOD | FINAL ACCURACY | Avg. STD |
|--------|----------------|----------|
| SGD | $93.302\% \pm 0.199\%$ | $0.928\%$ |
| M-SGD | $94.710\% \pm 0.169\%$ | $0.995\%$ |
| AM1-SGD-I | $94.675\% \pm 0.177\%$ | $\mathbf{0.587\%}$ |
| AM1-SGD-II | $94.619\% \pm 0.152\%$ | $\mathbf{0.314\%}$ |

Figure 3 & Table 1: Detailed data of the curves in Figure 2b. Best viewed in color.

**Learning rate scheduler issue**  We observed that when we use schedulers with a large decay factor and the momentum $\beta$ is too large for the task (e.g., 0.995 for the task of this section), there would be a performance drop after the learning rate reduction. We believe that it is caused by the different cardinalities of iterates being averaged in $\tilde{x}^+$, which leads to a false momentum. This issue is resolved by restarting the algorithm after each learning rate reduction inspired by (O'donoghue & Candes, 2015). We include more discussion and evidence in Appendix A.4.

### 3.1 AM2-SGD: A VARIANT WITH IDENTICAL ITERATIONS

---
**Algorithm 2** AM2-SGD

---
**Input:**  Initial guess $x_0$, amortization length $m$, a point table $\phi = [\phi_1 \quad \cdots \quad \phi_m] \in \mathbb{R}^{d \times m}$, learning rate $\eta$, momentum $\beta$, iteration number $K$.
**Initialize:**  $\phi_j^0 = x_0, \forall j \in [m]^*$. $\{j_k \mid j_k \in [m]\}_{k=0}^{K-1}$ is a sequence of uniformly random indexes.
  If Option II is used, $\bar{\phi}^0 = x_0$. {a running average for the point table $\phi$}
1: **for** $k = 0, \ldots, K-1$ **do**
2:   $\phi_{j_k}^{k+1} = x_k - \eta \cdot \nabla f_{i_k}(x_k)$ and keep other entries unchanged (i.e., $\phi_j^{k+1} = \phi_j^k$ for $j \neq j_k$).
3:   $x_{k+1} = \phi_{j_k}^{k+1} + \beta \cdot (\phi_{j_{k+1}}^{k+1} - \phi_{j_k}^k)$. {adding amortized momentum}
4:   **if** Option II **then** $\bar{\phi}^{k+1} = \bar{\phi}^k + \frac{1}{m} \cdot \left(\phi_{j_k}^{k+1} - \phi_{j_k}^k\right)$.
5: **end for**
**Output:**  Option I (not recommended): $x_K$, Option II: $\bar{\phi}^K$.       * $[m]$ denotes the set $\{1, \ldots, m\}$.

---

While enjoying an improved efficiency, AM1-SGD does not have identical iterations[7], which to some extent limits its extensibility to other settings (e.g., asynchronous setting). In this section, we propose a variant of Amortized Nesterov's Momentum (AM2-SGD, Algorithm 2) to address this problem. To show the characteristics of AM2-SGD, we make the following remarks:

**Trading memory for extensibility**  In expectation, the point table $\phi$ stores the most recent $m$ iterations and thus the output $\bar{\phi}^K$ is an $m$-iterations tail average, which connects to AM1-SGD. The relation between AM1-SGD and AM2-SGD resembles that of SVRG (Johnson & Zhang, 2013) and SAGA (Defazio et al., 2014), the most popular methods in finite-sum convex optimization: to reuse the information from several past iterates, we can either maintain a "snapshot" that aggregates the information or keep the iterates in a table. A side-by-side comparison is given in Section 4.

**Options and convergence**  As in the case of AM1-SGD, if $m = 1$, AM2-SGD with Option I corresponds to M-SGD and Option II is OM-SGD. In our preliminary experiments, the convergence of AM2-SGD is similar to AM1-SGD and it also has the learning rate scheduler issue. In our preliminary experiments (can be found in Appendix A), we observed that Option I is consistently worse than Option II and it does not seem to benefit from increasing $m$. Thus, we do not recommend using Option I. We also set $m = 5$ for AM2-SGD for its evaluation due to the similarity.

---
[7]For AM1-SGD, the workload varies for different iteration $k$ due to the if-clause at Step 4.

**Additional randomness** $\{j_k\}$    In our implementation, at each iteration, we sample an index in $[m]$ as $j_{k+1}$ and obtain the stored index $j_k$. We observed that with Option I, AM2-SGD has much larger deviations than AM1-SGD, which we believe is caused by the additional random indexes $\{j_k\}$.

## 4    CONVERGENCE RESULTS

The original Nesterov's accelerated gradient is famous for its optimal convergence rates for solving convex problems. In this section, we analyze the convergence rates for AM1-SGD and AM2-SGD in the convex case, which explicitly model the effect of amortization (i.e., $m$). While these rates do not hold for deep learning problems in general, they help us understand the observed convergence behaviors of the proposed methods, especially on how they differ from M-SGD ($m = 1$). Moreover, the analysis also provides intuition on tuning $m$. Since the original Nesterov's method is deterministic (Nesterov, 1983; 2013b), we follow the setting of its stochastic variants (Lan, 2012; Ghadimi & Lan, 2012), in which Nesterov's acceleration also achieves the optimal rates.

We consider the following convex composite problem (Beck & Teboulle, 2009; Nesterov, 2013a):

$$\min_{x \in X} \left\{ F(x) \triangleq f(x) + h(x) \right\}, \tag{4}$$

where $X \subseteq \mathbb{R}^d$ is a non-empty closed convex set and $h$ is a proper convex function with its proximal operator $\text{prox}_{\alpha h}(\cdot)$[8] available. We impose the following assumptions on the regularity of $f$ and the stochastic oracle $\nabla f_i$ (identical to the ones in Ghadimi & Lan (2012) with $\mu = 0$):

**Assumptions.** *For some $L \geq 0, M \geq 0, \sigma \geq 0$,*

*(a)* $0 \leq f(y) - f(x) - \langle \nabla f(x), y - x \rangle \leq \frac{L}{2} \|y - x\|^2 + M \|y - x\|, \forall x, y \in X.$[9]

*(b)* $\mathbb{E}_i [\nabla f_i(x)] = \nabla f(x), \forall x \in X.$

*(c)* $\mathbb{E}_i \big[ \|\nabla f_i(x) - \nabla f(x)\|^2 \big] \leq \sigma^2, \forall x \in X.$

The notation $\mathbb{E}_{i_k} [\,\cdot\,]$ is $\mathbb{E} [\,\cdot \mid (i_0, \ldots, i_{k-1})]$ for a random process $i_0, i_1, \ldots$. These assumptions cover several important classes of convex problems. For example, *(a)* covers the cases of $f$ being $L$-smooth ($M = 0$) or $L_0$-Lipschitz continuous ($M = 2L_0, L = 0$) convex functions and if $\sigma = 0$ in *(c)*, the assumptions cover several classes of deterministic convex programming problems. We denote $x^\star \in X$ as a solution to problem (4) and $x_0 \in X$ as the initial guess.

Unlike its usage in deep learning, the momentum parameter $\beta$ is always a variable in general convex analysis. For the simplicity of analysis, we reformulate AM1-SGD (Algorithm 1) and AM2-SGD (Algorithm 2) into the following schemes[10]($z \in X, \alpha \in \mathbb{R}$):

| **AM1-SGD** (reformulated, proximal) | **AM2-SGD** (reformulated, proximal) |
|---|---|
| **Initialize:** $\tilde{x}_0 = z_0 = x_0, S = K/m.$ | **Initialize:** $z_0 = \phi_j^0 = x_0, \forall j \in [m].$ |
| 1: **for** $s = 0, \ldots, S - 1$ **do** | 1: **for** $k = 0, \ldots, K - 1$ **do** |
| 2:     **for** $j = 0, \ldots, m - 1$ **do** | 2:     Sample $j_k$ uniformly in $[m]$. |
| 3:         $k = sm + j.$ | 3:     $x_k^{j_k} = (1 - \beta_k) \cdot z_k + \beta_k \cdot \phi_{j_k}^k.$ |
| 4:         $x_k = (1 - \beta_s) \cdot z_k + \beta_s \cdot \tilde{x}_s.$ | 4:     $z_{k+1} = \text{prox}_{\alpha_k h} \{z_k - \alpha_k \cdot \nabla f_{i_k}(x_k^{j_k})\}.$ |
| 5:         $z_{k+1} = \text{prox}_{\alpha_s h} \{z_k - \alpha_s \cdot \nabla f_{i_k}(x_k)\}.$ | 5:     $\phi_{j_k}^{k+1} = (1 - \beta_k) \cdot z_{k+1} + \beta_k \cdot \phi_{j_k}^k.$ |
| 6:         $(x_{k+1} = (1 - \beta_s) \cdot z_{k+1} + \beta_s \cdot \tilde{x}_s.)$ | 6: **end for** |
| 7:     **end for** | **Output:** $\bar{\phi}^K = \frac{1}{m} \sum_{j=1}^m \phi_j^K.$ |
| 8:     $\tilde{x}_{s+1} = \frac{1}{m} \sum_{j=1}^m x_{sm+j}.$ | |
| 9: **end for** | |
| **Output:** $\tilde{x}_S.$ | |

We show in Appendix B.1 that when $h \equiv 0$ and $\beta$ is a constant, the reformulated schemes AM1-SGD and AM2-SGD are equivalent to Algorithm 1 and Algorithm 2 through $\alpha_s = \eta(1 - \beta_s)^{-1}$ and

---

[8]$\forall x \in \mathbb{R}^d, \text{prox}_{\alpha h}(x) \triangleq \arg\min_{u \in X} \left\{ \frac{1}{2} \|u - x\|^2 + \alpha h(u) \right\}$, see Parikh et al. (2014).

[9]When $M > 0$, $f$ is not necessarily differentiable and we keep using the notation $\nabla f(x)$ to denote an arbitrary subgradient of $f$ at $x$ for consistency.

[10]For simplicity, we assume $K$ is divisible by $m$.

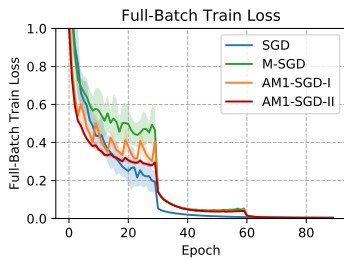 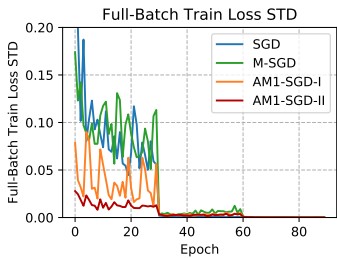

| METHOD | Avg. STD |
|---|---|
| SGD | 0.0329 |
| M-SGD | 0.0339 |
| AM1-SGD-I | **0.0142** |
| AM1-SGD-II | **0.0056** |

Figure 4 & Table 2: ResNet18 with pre-activation on CIFAR-10. For all methods, $\eta_0 = 0.1, \beta = 0.9$, run 20 seeds. For AM1-SGD, $m = 5$ and its labels are formatted as 'AM1-SGD-{*Option*}'. Shaded bands indicate $\pm 1$ standard deviation. Best viewed in color.

$\alpha_k = \eta(1 - \beta_k)^{-1}$. These reformulations are basically how Nesterov's momentum was migrated into deep learning (Sutskever et al., 2013). Then we establish the convergence rates for AM1-SGD and AM2-SGD as follows. All the proofs in this paper are given in Appendix B.2.

**Theorem 1.** *For the reformulated AM1-SGD, suppose we choose*

$$\beta_s = \frac{s}{s+2} \text{ and } \alpha_s = \frac{\lambda_1}{L(1-\beta_s)} \text{ with } \lambda_1 = \min\left\{\frac{2}{3}, \frac{L\|x_0 - x^\star\|}{2\sqrt{m}\sqrt{\sigma^2 + M^2}(S+1)^{\frac{3}{2}}}\right\}. \quad (5)$$

*Then,*

(a) *The output $\tilde{x}_S$ satisfies*

$$\mathbb{E}\left[F(\tilde{x}_S)\right] - F(x^\star) \le \frac{3Lm\|x_0 - x^\star\|^2}{(K+m)^2} + \frac{8\|x_0 - x^\star\|\sqrt{\sigma^2 + M^2}}{\sqrt{K+m}} \triangleq \mathcal{K}_0(m).$$

(b) *If the variance has a "light tail", i.e., $\mathbb{E}_i\left[\exp\left\{\|\nabla f_i(x) - \nabla f(x)\|^2/\sigma^2\right\}\right] \le \exp\{1\}, \forall x \in X$, and $X$ is compact, denoting $D_X \triangleq \max_{x \in X}\|x - x^\star\|$, for any $\Lambda \ge 0$, we have*

$$\text{Prob}\left\{F(\tilde{x}_S) - F(x^\star) \le \mathcal{K}_0(m) + \frac{4\Lambda\sigma\left(3\|x_0 - x^\star\| + \sqrt{6}D_X\right)}{3\sqrt{K+m}}\right\}$$
$$\ge 1 - \left(\exp\{-\Lambda^2/3\} + \exp\{-\Lambda\}\right).$$

*Remarks:* (a) Regarding $\mathcal{K}_0(m)$, its minimum is obtained at either $m = 1$ or $m = K$. Note that for AM1-SGD, $m$ is strictly constrained in $\{1, \ldots, K\}$. It can be verified that when $m = K$, AM1-SGD becomes the modified mirror descent SA (Lan, 2012), or under the Euclidean setting, the SGD that outputs the average of the whole history, which is rarely used in practice. In this case, the convergence rate in Theorem 1a becomes the corresponding $O(L/K + (\sigma + M)/\sqrt{K})$ (cf. Theorem 1 in Lan (2012)). Thus, we can regard AM1-SGD as a smooth transition between AC-SA and the modified mirror descent SA. (b) The additional compactness and "light tail" assumptions are similarly required in Nemirovski et al. (2009); Lan (2012); Ghadimi & Lan (2012). Recently, Juditsky et al. (2019) established similar bounds under weaker assumptions by truncating the gradient. However, as indicated by the authors, their technique cannot be used for accelerated algorithms due to the accumulation of bias.

*Understandings:* Theorem 1a gives the expected performance in terms of full-batch loss $F(\tilde{x}) - F(x^\star)$, from which the trade-off of $m$ is clear: Increasing $m$ improves the dependence on variance $\sigma$ but deteriorates the $O(L/K^2)$ term (i.e., the acceleration). Based on this trade-off, we can understand the empirical results in Figure 2b: the faster convergence in the early stage could be the result of a better control on $\sigma$ and the slightly lowered final accuracy is possibly caused by the reduced acceleration effect. Theorem 1b provides the probability of the full-batch loss deviating from its expected performance (i.e., $\mathcal{K}_0(m)$). It is clear that increasing $m$ leads to smaller deviations with the same probability, which sheds light on the understanding of the increased robustness observed in Figure 2. Since the theorem is built on the full-batch loss, we did an experiments based on this

metric in Figure 4 & Table 2. Here we choose training a smaller ResNet18 with pre-activation (He et al., 2016a) on CIFAR-10 as the case study (the test accuracy is reported in Appendix A.5).

For AM2-SGD, we only give the expected convergence results as follows.

**Theorem 2.** *For the reformulated AM2-SGD, if we choose*

$$\beta_k = \frac{k/m}{k/m+2} \text{ and } \alpha_k = \frac{\lambda_2}{L(1-\beta_k)} \text{ with } \lambda_2 = \min\left\{\frac{2}{3}, \frac{L\|x_0 - x^\star\|}{\sqrt{2m}(\sigma+M)\left(\frac{K-1}{m}+2\right)^{\frac{3}{2}}}\right\},$$

*the output $\bar{\phi}^K$ satisfies*

$$\mathbb{E}\left[F(\bar{\phi}^K)\right] - F(x^\star) \le \frac{4(m^2-m)\big(F(x_0)-F(x^\star)\big)+3Lm\|x_0-x^\star\|^2}{(K+2m-1)^2} + \frac{4\sqrt{2}\|x_0-x^\star\|(\sigma+M)}{\sqrt{K+2m-1}}.$$

*Remark:* In comparison with Theorem 1a, Theorem 2 has an additional term $F(x_0) - F(x^\star)$ in the upper bound, which is inevitable. This difference comes from different restrictions on the choice of $m$. For AM2-SGD, $m \ge 1$ is the only requirement. Since it is impossible to let $m \gg K$ to obtain an improved rate, this additional term is inevitable. As a sanity check, we can let $m \to \infty$ to obtain a point table with almost all $x_0$, and then the upper bound becomes exactly $F(x_0) - F(x^\star)$. In some cases, there exists an optimal choice of $m > 1$ in Theorem 2. However, the optimal choice could be messy and thus we omit the discussion here.

*Understanding:* Comparing the rates, we see that when using the same $m$, AM2-SGD has slightly better dependence on $\sigma$, which is related to the observation in Figure 5 that AM2-SGD is always slightly faster than AM1-SGD. This difference is suggesting that randomly incorporating past iterates beyond $m$ iterations helps.

If $m = O(1)$, Theorems 1 and 2 establish the optimal $O(L/K^2 + (\sigma + M)/\sqrt{K})$ rate in the convex setting (see Lan (2012) for optimality), which verifies AM1-SGD and AM2-SGD as variants of the Nesterov's method (Nesterov, 1983; 2013b). From the above analysis, the effect of $m$ can be understood as trading acceleration for variance control. However, since both acceleration and variance control boost the convergence speed, the reduced final performance observed in the CIFAR experiments may not always be the case as will be shown in Figure 5 and Table 3.

**Connections with Katyusha**    Our original inspiration of AM1-SGD comes from the construction of Katyusha (Allen-Zhu, 2018), the recent breakthrough in finite-sum convex optimization, which uses a previously calculated "snapshot" point to provide momentum, i.e., Katyusha momentum. AM1-SGD also uses an aggregated point to provide momentum and it shares many structural similarities with Katyusha. We refer the interested readers to Appendix B.3.

## 5    PERFORMANCE EVALUATION

In this section, we evaluate AM1-SGD and AM2-SGD on more deep learning tasks. Our goal is to show their potentials of serving as alternatives for M-SGD. Regarding the options: for AM1-SGD, Option I is a nice choice, which has slightly better final performance as shown in Table 1; for AM2-SGD, Option I is not recommended as mentioned before. Here we choose to evaluate Option II for both methods for consistency, which also corresponds to the analysis in Section 4. AM1-SGD and AM2-SGD use exactly the same values for $(\eta, \beta)$ as M-SGD, which was tuned to optimize the performance of M-SGD. We set $m = 5$ for AM1-SGD and AM2-SGD.

We trained ResNet50 and ResNet152 (He et al., 2016b) on the ILSVRC2012 dataset ("ImageNet") (Russakovsky et al., 2015) shown in Figure 5b. For this task, we used $0.1$ initial learning rate and $0.9$ momentum for all methods, which is a typical choice. We performed a restart after each learning rate reduction as discussed in Appendix A.4. We believe that this helps the training process and also does not incur any additional overhead. We report the final accuracy in Table 3.

We also did a language model experiment on Penn Treebank dataset (Marcus et al., 1993). We used the LSTM (Hochreiter & Schmidhuber, 1997) model defined in Merity et al. (2017) and followed the experimental setup in its released code. We only changed the learning rate and momentum in

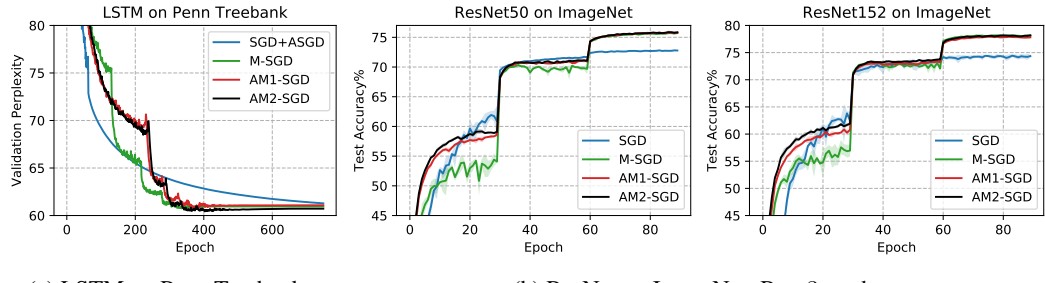

(a) LSTM on Penn Treebank.

(b) ResNet on ImageNet. Run 3 seeds.

Figure 5: Convergence of LSTM and ResNet. We plot the curve of validation perplexity and test accuracy, respectively. Shaded bands indicate $\pm 1$ standard deviation. Best viewed in color.

Table 3: Detailed perplexity and accuracy results for Figure 5.

| METHOD | Penn Treebank (Perplexity) | | ImageNet (Final Accuracy) | |
|---|---|---|---|---|
| | Validation | Test | ResNet50 | ResNet152 |
| SGD (+ASGD) | 61.283 | 59.068 | $72.783\% \pm 0.081\%$ | $74.361\% \pm 0.293\%$ |
| M-SGD | 60.747 | 58.355 | $75.711\% \pm 0.062\%$ | $78.065\% \pm 0.103\%$ |
| AM1-SGD | 60.734 | **57.977** | **75.779**$\% \pm 0.105\%$ | $77.816\% \pm 0.287\%$ |
| AM2-SGD | **60.434** | 58.233 | **75.845**$\% \pm 0.073\%$ | **78.194**$\% \pm 0.147\%$ |

the setup. The baseline is SGD+ASGD[11] (Polyak & Juditsky, 1992) with constant learning rate 30 as used in Merity et al. (2017). For the choice of $(\eta, \beta)$, following Lucas et al. (2019), we chose $\beta = 0.99$ and used the scheduler that reduces the learning rate by half when the validation loss has not decreased for 15 epochs. We swept $\eta$ from $\{5, 2.5, 1, 0.1, 0.01\}$ and found that $\eta = 2.5$ resulted in the lowest validation perplexity for M-SGD. We thus ran AM1-SGD and AM2-SGD with this $(\eta, \beta)$ and $m = 5$. Due to the small decay factor, we did not restart AM1-SGD and AM2-SGD after learning rate reductions. The validation perplexity curve is plotted in Figure 5a. We report validation perplexity and test perplexity in Table 3. This experiment is directly comparable with the one in Lucas et al. (2019).

Extra results are provided in the appendices for interested readers: the robustness when using large $\beta$ (Appendix A.2), a CIFAR-100 experiment (Appendix A.6) and comparison with classical momentum (Polyak, 1964), AggMo (Lucas et al., 2019) and QHM (Ma & Yarats, 2019) (Appendix A.3).

## 6 CONCLUSIONS

We presented Amortized Nesterov's Momentum, which is a special variant of Nesterov's momentum that utilizes several past iterates to provide the momentum. Based on this idea, we designed two different realizations, namely, AM1-SGD and AM2-SGD. Both of them are simple to implement with little-to-no additional tuning overhead over M-SGD. Our empirical results demonstrate that switching to AM1-SGD and AM2-SGD produces faster early convergence and comparable final generalization performance. AM1-SGD is lightweight and has more robust iterates than M-SGD, and thus can serve as a favorable alternative to M-SGD in large-scale deep learning tasks. AM2-SGD could be favorable for more restrictive tasks (e.g., asynchronous training) due to its extensibility and good performance. Both the methods are proved optimal in the convex case, just like M-SGD. Based on the intuition from convex analysis, the proposed methods are trading acceleration for variance control, which provides hints for the hyper-parameter tuning.

---

[11]SGD+ASGD is to run SGD and switch to averaged SGD (ASGD) when a threshold is met.

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

# Appendices

## A  EXTRA EXPERIMENTAL RESULTS

In this appendix, we provide more experimental results to further evaluate the Amortized Nesterov's Momentum. Table 4 shows the detailed data of the parameter sweep experiments, where the convergence curves of these results are given in Appendix A.1. In Appendix A.2, we compare the robustness of AM1-SGD and M-SGD on large momentum parameters. In Appendix A.3, we empirically compare the Amortized Nesterov's Momentum with classical momentum (Polyak, 1964), aggregated momentum (Lucas et al., 2019) and quasi-hyperbolic momentum (Ma & Yarats, 2019). We discuss the issues with learning rate schedulers in Appendix A.4. We report the test accuracy results of the ResNet18 experiment (in Section 4) in Appendix A.5. A CIFAR-100 experiment is provided in Appendix A.6. We also provide a sanity check for our implementation in Appendix A.7.

Table 4: Final test accuracy and average accuracy STD of training ResNet34 on CIFAR-10 over 5 runs (including the detailed data of the curves in Figure 1 and Figure 2a). For all the methods, $\eta_0 = 0.1, \beta = 0.9$. Multiple runs start with the same $x_0$.

| METHOD | DESCRIPTION | FINAL ACCURACY | Avg. STD |
|--------|-------------|----------------|----------|
| SGD | Standard Pytorch | $93.406\% \pm 0.149\%$ | 0.991% |
| M-SGD | Standard Pytorch | $94.606\% \pm 0.152\%$ | 1.040% |
| AM1-SGD | Option I, $m = 1$, sanity check | $94.672\% \pm 0.143\%$ | 0.912% |
| AM1-SGD | Option I, $m = 3$ | $94.630\% \pm 0.025\%$ | 0.643% |
| AM1-SGD | Option I, $m = 5$ | $94.604\% \pm 0.104\%$ | 0.496% |
| AM1-SGD | Option I, $m = 7$ | $94.640\% \pm 0.127\%$ | 0.439% |
| AM1-SGD | Option I, $m = 10$ | $94.542\% \pm 0.133\%$ | 0.441% |
| AM1-SGD | Option I, $m = 20$ | $94.378\% \pm 0.217\%$ | 0.399% |
| AM1-SGD | Option I, $m = 30$ | $94.300\% \pm 0.145\%$ | 0.429% |
| OM-SGD | AM1-SGD (Opt. II, $m = 1$) | $94.728\% \pm 0.111\%$ | 0.634% |
| AM1-SGD | Option II, $m = 3$ | $94.662\% \pm 0.139\%$ | 0.411% |
| AM1-SGD | Option II, $m = 5$ | $94.602\% \pm 0.083\%$ | 0.265% |
| AM1-SGD | Option II, $m = 7$ | $94.514\% \pm 0.103\%$ | 0.282% |
| AM1-SGD | Option II, $m = 10$ | $94.424\% \pm 0.117\%$ | 0.293% |
| AM1-SGD | Option II, $m = 20$ | $94.358\% \pm 0.181\%$ | 0.306% |
| AM1-SGD | Option II, $m = 30$ | $94.270\% \pm 0.134\%$ | 0.344% |
| AM2-SGD | Option I, $m = 1$, sanity check | $94.682\% \pm 0.212\%$ | 0.822% |
| AM2-SGD | Option I, $m = 5$ | $94.572\% \pm 0.188\%$ | 0.591% |
| AM2-SGD | Option I, $m = 10$ | $94.440\% \pm 0.138\%$ | 0.737% |
| AM2-SGD | Option I, $m = 20$ | $94.312\% \pm 0.149\%$ | 0.741% |
| AM2-SGD | Option II, $m = 5$ | $94.664\% \pm 0.107\%$ | 0.263% |
| AM2-SGD | Option II, $m = 10$ | $94.496\% \pm 0.211\%$ | 0.280% |
| AM2-SGD | Option II, $m = 20$ | $94.412\% \pm 0.140\%$ | 0.246% |

### A.1  THE EFFECT OF $m$ ON CONVERGENCE

We show in Figure 6 how $m$ affects the convergence of test accuracy. The results show that increasing $m$ speeds up the convergence in the early stage. While for AM1-SGD the convergences of Option I and Option II are similar, AM2-SGD with Option II is consistently better than with Option I in this experiment. It seems that AM2-SGD with Option I does not benefit from increasing $m$ and the algorithm is not robust. Thus, we do not recommend using Option I for AM2-SGD.

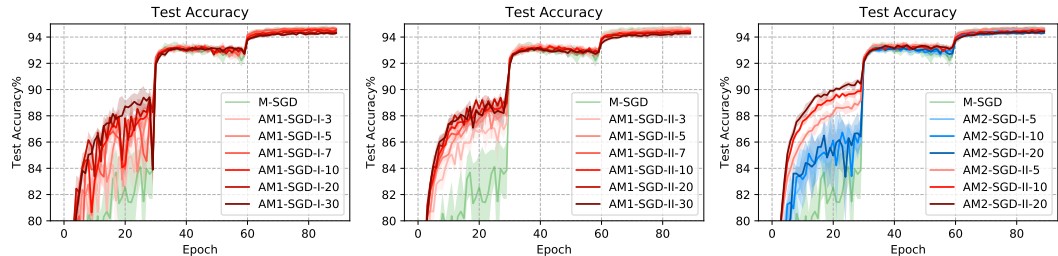

Figure 6: Convergence of test accuracy from the parameter sweep experiments in Table 4. Labels are formatted as 'AM1/2-SGD-{*Option*}-{$m$}' . Best viewed in color.

## A.2 ROBUSTNESS ON LARGE MOMENTUM PARAMETERS

We compare the robustness of M-SGD and AM1-SGD when $\beta$ is large in Figure 7 & Table 5. For fair comparison, AM1-SGD uses Option I. As we can see, the STD error of M-SGD scales up significantly when $\beta$ is larger and the performance is more affected by a large $\beta$ compared with AM1-SGD.

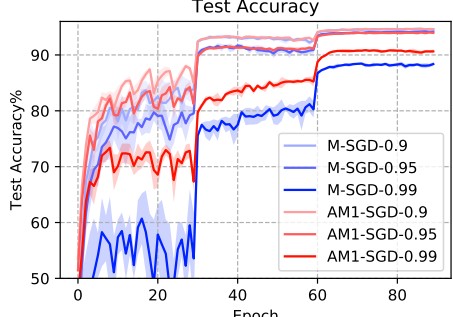

| METHOD | FINAL ACCURACY | Avg. STD |
|---|---|---|
| M-SGD-0.9 | $94.606\% \pm 0.152\%$ | 1.040% |
| M-SGD-0.95 | $94.198\% \pm 0.120\%$ | 1.200% |
| M-SGD-0.99 | $88.366\% \pm 0.357\%$ | 2.561% |
| AM1-SGD-0.9 | $94.604\% \pm 0.104\%$ | 0.496% |
| AM1-SGD-0.95 | $93.942\% \pm 0.074\%$ | 0.581% |
| AM1-SGD-0.99 | $90.640\% \pm 0.375\%$ | 0.900% |

Figure 7 & Table 5: ResNet34 on CIFAR-10. $\eta_0 = 0.1, \beta \in \{0.9, 0.95, 0.99\}$, run 5 seeds (the $\beta = 0.9$ results are copied from Table 4). Labels are formatted as "{*Algorithm*}-{$\beta$}". Best viewed in color.

## A.3 COMPARISON WITH OTHER MOMENTUM

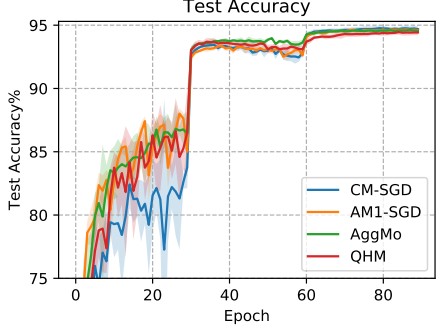

| METHOD | FINAL ACCURACY | Avg. STD |
|---|---|---|
| CM-SGD | $94.690\% \pm 0.212\%$ | 1.107% |
| AM1-SGD | $94.604\% \pm 0.104\%$ | 0.496% |
| AggMo | $94.556\% \pm 0.142\%$ | 0.777% |
| QHM | $94.426\% \pm 0.226\%$ | 1.073% |
| M-SGD | $94.606\% \pm 0.152\%$ | 1.040% |

Figure 8 & Table 6: ResNet34 on CIFAR-10. Run 5 seeds. The results of AM1-SGD and M-SGD are copied from Table 4. Best viewed in color.

In this section, we compare AM1-SGD (Option I) with classical momentum (Polyak, 1964), AggMo (Lucas et al., 2019) and QHM (Ma & Yarats, 2019) in our basic case study (training ResNet34 on

CIFAR-10). Since we are not aware of what makes a fair comparison with these methods (e.g., it is not clear what is the effective learning rate for AM1-SGD), we compare them based on the default hyper-parameter settings suggested by their papers.

**Classical Momentum**    The SGD with classical momentum (CM-SGD) that is widely used in deep learning has the following scheme (standard PyTorch) ($v^{cm} \in \mathbb{R}^d, v_0^{cm} = \mathbf{0}$):

$$v_{k+1}^{cm} = \beta \cdot v_k^{cm} + \nabla f_{i_k}(x_k),$$
$$x_{k+1} = x_k - \eta \cdot v_{k+1}^{cm}, \text{ for } k \geq 0.$$

CM-SGD with its typical hyper-parameter settings ($\eta_0 = 0.1, \beta = 0.9$) is observed to achieve similar generalization performance as M-SGD. However, CM-SGD is more unstable and prone to oscillations (Lucas et al., 2019), which makes it less robust than M-SGD as shown in Table 6.

**Aggregated Momentum (AggMo)**    AggMo combines multiple momentum buffers, which is inspired by the passive damping from physics literature (Lucas et al., 2019). AggMo uses the following update rules (for $t = 1, \ldots, T, v^{(t)} \in \mathbb{R}^d, v_0^{(t)} = \mathbf{0}$):

$$v_{k+1}^{(t)} = \beta^{(t)} \cdot v_k^{(t)} - \nabla f_{i_k}(x_k), \text{ for } t = 1, \ldots, T,$$
$$x_{k+1} = x_k + \frac{\eta}{T} \cdot \sum_{t=1}^{T} v_{k+1}^{(t)}, \text{ for } k \geq 0.$$

We used the exponential hyper-parameter setting recommended in the original work with the scale-factor $a = 0.1$ fixed, $\beta^{(t)} = 1 - a^{t-1}$, for $t = 1, \ldots, T$ and choosing $T$ in $\{2, 3, 4\}$. We found that $T = 2$ gave the best performance in this experiment. As shown in Figure 8 & Table 6, with the help of passive damping, AggMo is more stable and robust compared with CM-SGD.

**Quasi-hyperbolic Momentum (QHM)**    Ma & Yarats (2019) introduce the immediate discount factor $\nu \in \mathbb{R}$ for the momentum scheme, which results in the QHM update rules ($\alpha \in \mathbb{R}, v^{qh} \in \mathbb{R}^d, v_0^{qh} = \mathbf{0}$):

$$v_{k+1}^{qh} = \beta \cdot v_k^{qh} + (1 - \beta) \cdot \nabla f_{i_k}(x_k),$$
$$x_{k+1} = x_k - \alpha \cdot (\nu \cdot v_{k+1}^{qh} + (1 - \nu) \cdot \nabla f_{i_k}(x_k)), \text{ for } k \geq 0.$$

Here we used the recommended hyper-parameter setting for QHM ($\alpha_0 = 1.0, \beta = 0.999, \nu = 0.7$).

Figure 8 shows that AM1-SGD, AggMo and QHM achieve faster convergence in the early stage while CM-SGD has the highest final accuracy. In terms of robustness, huge gaps are observed when comparing AM1-SGD with the remaining methods in Table 6. Note that AM1-SGD is more efficient than both QHM and AggMo, and is as efficient as CM-SGD.

We also plot the convergence of train-batch loss for all the methods in Figure 9. Despite of showing worse generalization performance, both QHM and AggMo perform better on reducing the train-batch loss in this experiment, which is consistent with the results reported in Ma & Yarats (2019); Lucas et al. (2019).

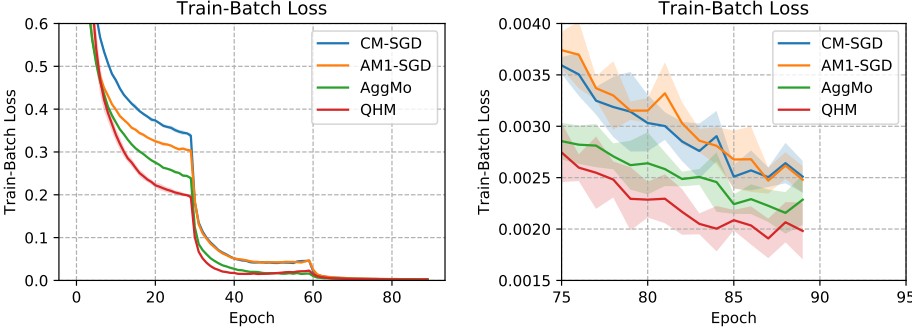

Figure 9: Train-batch loss results. Best viewed in color.

### A.4 ISSUES WITH LEARNING RATE SCHEDULERS

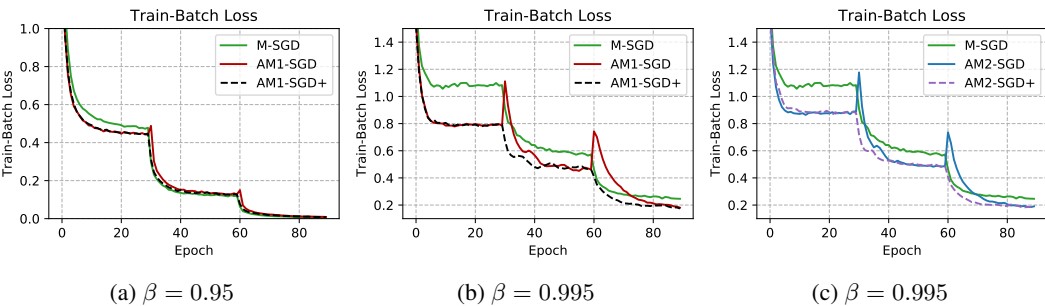

(a) $\beta = 0.95$        (b) $\beta = 0.995$        (c) $\beta = 0.995$

Figure 10: ResNet18 on CIFAR-10. $\eta_0 = 0.1, \beta \in \{0.95, 0.995\}$. '+' represents performing a restart after each learning rate reduction.

We show in Figure 10 that when $\beta$ is large for the task, using step learning rate scheduler with decay factor 10, a performance drop is observed after each reduction. Both Option I and Option II have this issue and the curves are basically identical. Here we only use Option II. We fix this issue by performing a restart after each learning rate reduction (labeled with '+'). We plot the train-batch loss here because we find the phenomenon is clearer in this way. If $\beta = 0.9$, there is no observable performance drop in this experiment.

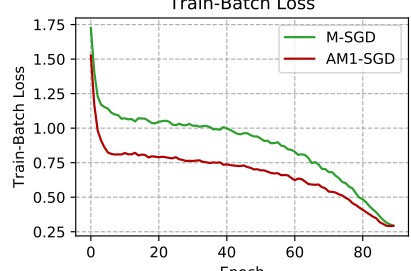

For smooth-changing schedulers such as the cosine annealing scheduler (Loshchilov & Hutter, 2016), the amortized momentum works well as shown in Figure 11.

Figure 11: ResNet18 on CIFAR-10. Cosine annealing scheduler (without restarts), $\eta_0 = 0.1, \beta = 0.995$.

### A.5 TEST ACCURACY RESULTS OF FIGURE 4 & TABLE 2

We report the test accuracy results of the experiments in Section 4 in Figure 12 & Table 7. These results are reminiscent of the ResNet34 experiments (Figure 3 & Table 1).

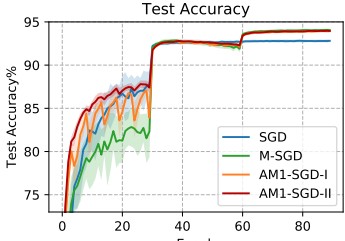

| METHOD | FINAL ACCURACY | Avg. STD |
|--------|----------------|----------|
| SGD | $92.810\% \pm 0.147\%$ | $1.005\%$ |
| M-SGD | $94.057\% \pm 0.170\%$ | $1.102\%$ |
| AM1-SGD-I | $93.968\% \pm 0.154\%$ | **$0.543\%$** |
| AM1-SGD-II | $93.953\% \pm 0.192\%$ | **$0.315\%$** |

Figure 12 & Table 7: ResNet18 with pre-activation on CIFAR-10. For all methods, $\eta_0 = 0.1, \beta = 0.9$, run 20 seeds. For AM1-SGD, $m = 5$ and its labels are formatted as 'AM1-SGD-{*Option*}'. Shaded bands indicate $\pm 1$ standard deviation. Best viewed in color.

### A.6 CIFAR-100 EXPERIMENT

We report the results of training DenseNet121 (Huang et al., 2017) on CIFAR-100 in Figure 13, which shows that both AM1-SGD and AM2-SGD perform well before the final learning rate reduction. However, the final accuracies are lowered around $0.6\%$ compared with M-SGD. We also notice that SGD reduces the train-batch loss at an incredibly fast rate and the losses it reaches are consistently lower than other methods in the entire 300 epochs. However, this performance is not

reflected in the convergence of test accuracy. We believe that this phenomenon suggests that the DenseNet model is actually "overfitting" M-SGD (since in the ResNet experiments, M-SGD always achieves a lower train loss than SGD after the final learning rate reduction).

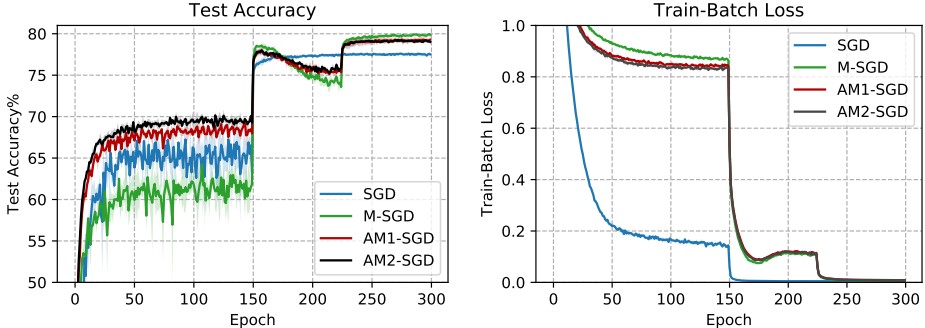

Figure 13: DenseNet121 on CIFAR-100. For all methods, $\eta_0 = 0.1, \beta = 0.9$, run 3 seeds. AM1-SGD and AM2-SGD use Option II and $m = 5$. Shaded bands indicating $\pm 1$ standard deviation. Best viewed in color.

### A.7 A SANITY CHECK

When $m = 1$, both AM1-SGD and AM2-SGD are equivalent to M-SGD, we plot their convergence in Figure 14 as a sanity check (the detailed data is given in Table 4).

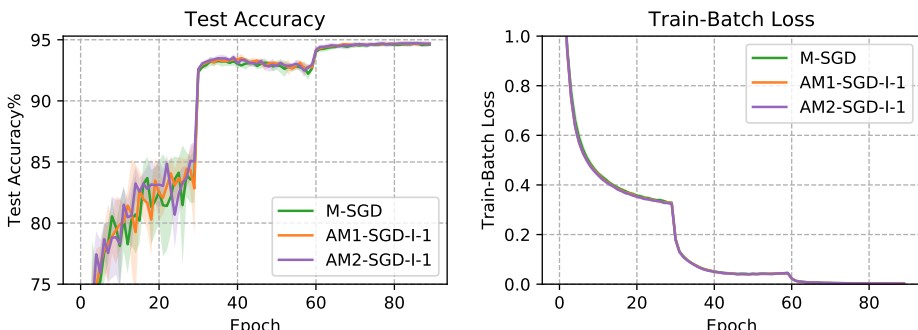

Figure 14: A sanity check. Labels are formatted as 'AM{1/2}-SGD-{*Option*}-{*m*}'.

We observed that when $m = 1$, both AM1-SGD and AM2-SGD have a lower STD error than M-SGD. We believe that it is because they both maintain the iterates without scaling, which is numerically more stable than M-SGD (M-SGD in standard PyTorch maintains a scaled buffer, i.e., $v_k^{pt} = \eta^{-1}\beta^{-1} \cdot (y_k - x_k)$).

## B  MISSING PARTS IN SECTION 4

### B.1  THE REFORMULATIONS

When $h \equiv 0$ and $\beta$ is a constant, we do the reformulations by eliminating the sequence $\{z_k\}$.

For the reformulated AM2-SGD,

$$
\begin{aligned}
x_k^{j_k} &= (1 - \beta) \cdot z_k + \beta \cdot \phi_{j_k}^k, \\
z_{k+1} &= z_k - \alpha \cdot \nabla f_{i_k}(x_k^{j_k}), \\
\phi_{j_k}^{k+1} &= (1 - \beta) \cdot z_{k+1} + \beta \cdot \phi_{j_k}^k, \\
\left( x_{k+1}^{j_{k+1}} \right. &= \left. (1 - \beta) \cdot z_{k+1} + \beta \cdot \phi_{j_{k+1}}^{k+1} \right).
\end{aligned}
\qquad
\begin{aligned}
&\alpha(1 - \beta) = \eta \\
&\underline{\text{Eliminate } \{z_k\}} \\
&\xRightarrow{\hspace{2cm}}
\end{aligned}
\qquad
\begin{aligned}
\phi_{j_k}^{k+1} &= x_k^{j_k} - \eta \cdot \nabla f_{i_k}(x_k^{j_k}), \\
x_{k+1}^{j_{k+1}} &= \phi_{j_k}^{k+1} + \beta \cdot \left( \phi_{j_{k+1}}^{k+1} - \phi_{j_k}^k \right).
\end{aligned}
$$


The reformulated AM2-SGD         Algorithm 2


For the reformulated AM1-SGD, when $h \equiv 0$, the inner loops are basically SGD,

$$
\begin{aligned}
x_k &= (1 - \beta) \cdot z_k + \beta \cdot \tilde{x}_s, \\
z_{k+1} &= z_k - \alpha \cdot \nabla f_{i_k}(x_k), \\
(x_{k+1} &= (1 - \beta) \cdot z_{k+1} + \beta \cdot \tilde{x}_s.)
\end{aligned}
\qquad
\begin{aligned}
&\alpha(1 - \beta) = \eta \\
&\underline{\text{Eliminate } \{z_k\}} \\
&\xRightarrow{\hspace{2cm}}
\end{aligned}
\qquad
x_{k+1} = x_k - \eta \cdot \nabla f_{i_k}(x_k).
$$

At the end of each inner loop (i.e., when $(k+1) \mod m = 0$), we have

$$
x_{(s+1)m} = (1 - \beta) \cdot z_{(s+1)m} + \beta \cdot \tilde{x}_s,
$$

while at the beginning of the next inner loop,

$$
x_{(s+1)m} = (1 - \beta) \cdot z_{(s+1)m} + \beta \cdot \tilde{x}_{s+1},
$$

which means that we need to set $x_{k+1} \leftarrow x_{k+1} + \beta \cdot (\tilde{x}_{s+1} - \tilde{x}_s)$ (reassign the value of $x_{k+1}$).

We also give the reformulation of M-SGD (scheme (1)) to the Auslender & Teboulle (2006) scheme for reference:

$$
\begin{aligned}
x_k &= (1 - \beta) \cdot z_k + \beta \cdot y_k, \\
z_{k+1} &= z_k - \alpha \cdot \nabla f_{i_k}(x_k), \\
y_{k+1} &= (1 - \beta) \cdot z_{k+1} + \beta \cdot y_k, \\
\left( x_{k+1} \right. &= \left. (1 - \beta) \cdot z_{k+1} + \beta \cdot y_{k+1} \right).
\end{aligned}
\qquad
\begin{aligned}
&\alpha(1 - \beta) = \eta \\
&\underline{\text{Eliminate } \{z_k\}} \\
&\xRightarrow{\hspace{2cm}}
\end{aligned}
\qquad
\begin{aligned}
y_{k+1} &= x_k - \eta \cdot \nabla f_{i_k}(x_k), \\
x_{k+1} &= y_{k+1} + \beta \cdot (y_{k+1} - y_k).
\end{aligned}
$$


Auslender & Teboulle (2006)
(AC-SA (Lan, 2012))         Nesterov (1983; 2013b)


AC-SA (in the Euclidean case) maps to the Auslender & Teboulle (2006) scheme through (in the original notations)

$$
\begin{cases}
x = x^{md} \\
z = x \\
y = x^{ag} \\
1 - \beta = \beta_t^{-1} \\
\alpha = \gamma_t
\end{cases}
.
$$

Intuition for the Auslender & Teboulle (2006) scheme can be found in Remark 2 in Lan (2012).

B.2 Proofs of Theorem 1 and Theorem 2

The reformulated schemes are copied here for reference:

**AM1-SGD** (reformulated, proximal)  | **AM2-SGD** (reformulated, proximal)

**AM1-SGD** (reformulated, proximal)

**Initialize:** $\tilde{x}_0 = z_0 = x_0, S = K/m.$
1: **for** $s = 0, \ldots, S - 1$ **do**
2:   **for** $j = 0, \ldots, m - 1$ **do**
3:     $k = sm + j.$
4:     $x_k = (1 - \beta_s) \cdot z_k + \beta_s \cdot \tilde{x}_s.$
5:     $z_{k+1} = \text{prox}_{\alpha_s h}\{z_k - \alpha_s \cdot \nabla f_{i_k}(x_k)\}.$
6:     $(x_{k+1} = (1 - \beta_s) \cdot z_{k+1} + \beta_s \cdot \tilde{x}_s.)$
7:   **end for**
8:   $\tilde{x}_{s+1} = \frac{1}{m}\sum_{j=1}^{m} x_{sm+j}.$
9: **end for**
**Output:** $\tilde{x}_S.$

**AM2-SGD** (reformulated, proximal)

**Initialize:** $z_0 = \phi_j^0 = x_0, \forall j \in [m].$
1: **for** $k = 0, \ldots, K - 1$ **do**
2:   Sample $j_k$ uniformly in $[m].$
3:   $x_k^{j_k} = (1 - \beta_k) \cdot z_k + \beta_k \cdot \phi_{j_k}^k.$
4:   $z_{k+1} = \text{prox}_{\alpha_k h}\{z_k - \alpha_k \cdot \nabla f_{i_k}(x_k^{j_k})\}.$
5:   $\phi_{j_k}^{k+1} = (1 - \beta_k) \cdot z_{k+1} + \beta_k \cdot \phi_{j_k}^k.$
6: **end for**
**Output:** $\bar{\phi}^K = \frac{1}{m}\sum_{j=1}^{m} \phi_j^K.$

Comparing the reformulated schemes, we see that their iterations can be generalized as follows:

$$
\begin{aligned}
x &= (1 - \beta) \cdot z + \beta \cdot y, \\
z^+ &= \text{prox}_{\alpha h}\{z - \alpha \cdot \nabla f_i(x)\}, \\
y^+ &= (1 - \beta) \cdot z^+ + \beta \cdot y.
\end{aligned}
\tag{6}
$$

This type of scheme is first proposed in Auslender & Teboulle (2006), which represents one of the simplest variants of the Nesterov's methods (see Tseng (2008) for other variants). The scheme is then modified into various settings (Hu et al., 2009; Lan, 2012; Ghadimi & Lan, 2012; 2016; Zhou et al., 2019; Lan et al., 2019) to achieve acceleration. The following lemma serves as a cornerstone for the convergence proofs of AM1-SGD and AM2-SGD.

**Lemma 1.** *If $\alpha(1 - \beta) < 1/L$, the update scheme (6) satisfies the following recursion:*

$$
\begin{aligned}
\frac{1}{1 - \beta}\big(F(y^+) - F(x^\star)\big) &\le \frac{\beta}{1 - \beta}\big(F(y) - F(x^\star)\big) + \frac{1}{2\alpha}\left(\|z - x^\star\|^2 - \|z^+ - x^\star\|^2\right) \\
&+ \frac{(\|\nabla f(x) - \nabla f_i(x)\| + M)^2}{2(\alpha^{-1} - L(1 - \beta))} + \langle \nabla f(x) - \nabla f_i(x), z - x^\star \rangle.
\end{aligned}
$$

B.2.1 Proof of Lemma 1

This Lemma is similarly provided in Lan (2012); Ghadimi & Lan (2012) under a more general setting that allows non-Euclidean norms in the assumptions, we give a proof here for completeness.

Based on the convexity (Assumption (a)), we have

$$
\begin{aligned}
f(x) - f(x^\star) &\le \underbrace{\langle \nabla f(x), x - z \rangle}_{R_0} + \underbrace{\langle \nabla f(x) - \nabla f_i(x), z - x^\star \rangle}_{R_1} + \underbrace{\langle \nabla f_i(x), z - z^+ \rangle}_{R_2} \\
&+ \underbrace{\langle \nabla f_i(x), z^+ - x^\star \rangle}_{R_3}.
\end{aligned}
\tag{7}
$$

We upper bound the terms on the right side one-by-one.

For $R_0$,

$$
R_0 \overset{(\star)}{=} \frac{\beta}{1 - \beta}\langle \nabla f(x), y - x \rangle \le \frac{\beta}{1 - \beta}\big(f(y) - f(x)\big),
\tag{8}
$$

where $(\star)$ uses the relation between $x$ and $z$, i.e., $(1 - \beta) \cdot (x - z) = \beta \cdot (y - x).$

For $R_2$, based on Assumption (a), we have

$$
f(y^+) - f(x) + \langle \nabla f(x), x - y^+ \rangle \le \frac{L}{2}\|x - y^+\|^2 + M\|x - y^+\|.
$$

Then, noting that $x - y^+ = (1 - \beta) \cdot (z - z^+)$, we can arrange the above inequality as

$$
\begin{aligned}
R_2 &\leq \frac{L(1-\beta)}{2} \left\| z - z^+ \right\|^2 + \frac{1}{1-\beta} \left( f(x) - f(y^+) \right) + \left\langle \nabla f(x) - \nabla f_i(x), z^+ - z \right\rangle \\
&\quad + M \left\| z - z^+ \right\| \\
&\leq \frac{L(1-\beta)}{2} \left\| z - z^+ \right\|^2 + \frac{1}{1-\beta} \left( f(x) - f(y^+) \right) + \left( \left\| \nabla f(x) - \nabla f_i(x) \right\| + M \right) \left\| z - z^+ \right\|.
\end{aligned}
$$

Using Young's inequality with $\zeta > 0$, we obtain

$$
R_2 \leq \frac{L(1-\beta) + \zeta}{2} \left\| z - z^+ \right\|^2 + \frac{1}{1-\beta} \left( f(x) - f(y^+) \right) + \frac{\left( \left\| \nabla f(x) - \nabla f_i(x) \right\| + M \right)^2}{2\zeta}. \quad (9)
$$

For $R_3$, based on the optimality condition of $\mathrm{prox}_{\alpha h}\{z - \alpha \cdot \nabla f_i(x)\}$ and denoting $\partial h(z^+)$ as a subgradient of $h$ at $z^+$, we have for any $u \in X$,

$$
\left\langle \alpha \cdot \partial h(z^+) + z^+ - z + \alpha \cdot \nabla f_i(x), u - z^+ \right\rangle \geq 0,
$$

$$
\left\langle \nabla f_i(x), z^+ - u \right\rangle \leq \left\langle \partial h(z^+), u - z^+ \right\rangle + \frac{1}{\alpha} \left\langle z^+ - z, u - z^+ \right\rangle
$$

$$
\leq h(u) - h(z^+) + \frac{1}{\alpha} \left\langle z^+ - z, u - z^+ \right\rangle.
$$

Choosing $u = x^\star$,

$$
\begin{aligned}
R_3 &\leq h(x^\star) - h(z^+) + \frac{1}{\alpha} \left\langle z^+ - z, x^\star - z^+ \right\rangle \\
&\overset{(\star)}{=} h(x^\star) - h(z^+) + \frac{1}{2\alpha} \left( \left\| z - x^\star \right\|^2 - \left\| z^+ - x^\star \right\|^2 - \left\| z^+ - z \right\|^2 \right), \quad (10)
\end{aligned}
$$

where $(\star)$ follows from $\|a + b\|^2 = \|a\|^2 + \|b\|^2 + 2\langle a, b \rangle$.

Finally, by upper bounding (7) using (8), (9), (10), we conclude that

$$
\begin{aligned}
f(x) - f(x^\star) &\leq R_1 + \frac{\beta}{1-\beta} \left( f(y) - f(x) \right) + \frac{L(1-\beta) + \zeta - \alpha^{-1}}{2} \left\| z - z^+ \right\|^2 \\
&\quad + \frac{1}{1-\beta} \left( f(x) - f(y^+) \right) + h(x^\star) - h(z^+) + \frac{\left( \left\| \nabla f(x) - \nabla f_i(x) \right\| + M \right)^2}{2\zeta} \\
&\quad + \frac{1}{2\alpha} \left( \left\| z - x^\star \right\|^2 - \left\| z^+ - x^\star \right\|^2 \right),
\end{aligned}
$$

After simplification,

$$
\begin{aligned}
\frac{1}{1-\beta} \left( f(y^+) - f(x^\star) \right) &\leq \frac{\beta}{1-\beta} \left( f(y) - f(x^\star) \right) + \frac{L(1-\beta) + \zeta - \alpha^{-1}}{2} \left\| z - z^+ \right\|^2 \\
&\quad + h(x^\star) - h(z^+) + \frac{\left( \left\| \nabla f(x) - \nabla f_i(x) \right\| + M \right)^2}{2\zeta} + R_1 \quad (11) \\
&\quad + \frac{1}{2\alpha} \left( \left\| z - x^\star \right\|^2 - \left\| z^+ - x^\star \right\|^2 \right).
\end{aligned}
$$

Note that with the convexity of $h$ and $y^+ = (1 - \beta) \cdot z^+ + \beta \cdot y$, we have

$$
h(y^+) \leq (1-\beta)h(z^+) + \beta h(y),
$$

$$
h(z^+) \geq \frac{1}{1-\beta} h(y^+) - \frac{\beta}{1-\beta} h(y).
$$

Using the above inequality and choosing $\zeta = \alpha^{-1} - L(1-\beta) > 0 \Rightarrow \alpha(1-\beta) < 1/L$, we can arrange (11) as

$$
\begin{aligned}
\frac{1}{1-\beta} \left( F(y^+) - F(x^\star) \right) &\leq \frac{\beta}{1-\beta} \left( F(y) - F(x^\star) \right) + \frac{1}{2\alpha} \left( \left\| z - x^\star \right\|^2 - \left\| z^+ - x^\star \right\|^2 \right) \\
&\quad + \frac{\left( \left\| \nabla f(x) - \nabla f_i(x) \right\| + M \right)^2}{2(\alpha^{-1} - L(1-\beta))} + R_1.
\end{aligned}
$$

### B.2.2 Proof of Theorem 1A

Using Assumption (c), Lemma 1 with

$$
\begin{cases}
x = x_k \\
z = z_k \\
z^+ = z_{k+1} \\
y = \tilde{x}_s \\
y^+ = x_{k+1} \\
\alpha = \alpha_s \\
\beta = \beta_s
\end{cases}
, \tag{12}
$$

and taking expectation, if $\alpha_s(1 - \beta_s) < 1/L$, we have

$$
\frac{1}{1 - \beta_s} \left( \mathbb{E}_{i_k} [F(x_{k+1})] - F(x^\star) \right) + \frac{1}{2\alpha_s} \mathbb{E}_{i_k} \left[ \|z_{k+1} - x^\star\|^2 \right]
$$
$$
\leq \frac{\beta_s}{1 - \beta_s} \left( F(\tilde{x}_s) - F(x^\star) \right) + \frac{1}{2\alpha_s} \|z_k - x^\star\|^2 + \frac{(\sigma + M)^2}{2(\alpha_s^{-1} - L(1 - \beta_s))}.
$$

Summing the above inequality from $k = sm, \dots, sm + m - 1$, we obtain

$$
\frac{1}{(1 - \beta_s)m} \sum_{j=1}^{m} \left( \mathbb{E} [F(x_{sm+j})] - F(x^\star) \right) + \frac{1}{2\alpha_s m} \mathbb{E} \left[ \|z_{(s+1)m} - x^\star\|^2 \right]
$$
$$
\leq \frac{\beta_s}{1 - \beta_s} \left( F(\tilde{x}_s) - F(x^\star) \right) + \frac{1}{2\alpha_s m} \|z_{sm} - x^\star\|^2 + \frac{(\sigma + M)^2}{2(\alpha_s^{-1} - L(1 - \beta_s))},
$$

Using the definition of $\tilde{x}_{s+1}$ and convexity,

$$
\frac{\alpha_s}{1 - \beta_s} \left( \mathbb{E} [F(\tilde{x}_{s+1})] - F(x^\star) \right) + \frac{1}{2m} \mathbb{E} \left[ \|z_{(s+1)m} - x^\star\|^2 \right]
$$
$$
\leq \frac{\alpha_s \beta_s}{1 - \beta_s} \left( F(\tilde{x}_s) - F(x^\star) \right) + \frac{1}{2m} \|z_{sm} - x^\star\|^2 + \frac{\alpha_s(\sigma^2 + M^2)}{\alpha_s^{-1} - L(1 - \beta_s)}. \tag{13}
$$

It can be verified that with the choices $\beta_s = \frac{s}{s+2}$ and $\alpha_s = \frac{\lambda_1}{L(1 - \beta_s)}$, the following holds for $s \geq 0$,

$$
\frac{\alpha_{s+1}\beta_{s+1}}{1 - \beta_{s+1}} \leq \frac{\alpha_s}{1 - \beta_s} \text{ and } \beta_0 = 0. \tag{14}
$$

Note that since our analysis aims at providing intuition, we do not refine the choice of $\alpha_s$ as in (Hu et al., 2009; Ghadimi & Lan, 2012). Thus, by telescoping (13) from $s = S - 1, \dots, 0$, we obtain

$$
\frac{\alpha_{S-1}}{1 - \beta_{S-1}} \left( \mathbb{E} [F(\tilde{x}_S)] - F(x^\star) \right) + \frac{1}{2m} \mathbb{E} \left[ \|z_{Sm} - x^\star\|^2 \right]
$$
$$
\leq \frac{1}{2m} \|x_0 - x^\star\|^2 + \sum_{s=0}^{S-1} \frac{\alpha_s(\sigma^2 + M^2)}{\alpha_s^{-1} - L(1 - \beta_s)},
$$

and thus,

$$
\mathbb{E} [F(\tilde{x}_S)] - F(x^\star) \leq \frac{2L}{\lambda_1 m (S+1)^2} \|x_0 - x^\star\|^2 + \frac{4L(\sigma^2 + M^2)}{\lambda_1 (S+1)^2} \sum_{s=0}^{S-1} \frac{\alpha_s^2}{1 - \alpha_s(1 - \beta_s)L}
$$
$$
\overset{(a)}{\leq} \frac{2L}{\lambda_1 m (S+1)^2} \|x_0 - x^\star\|^2 + \frac{3\lambda_1(\sigma^2 + M^2)}{L(S+1)^2} \sum_{s=0}^{S-1} (s+2)^2
$$
$$
\overset{(b)}{\leq} \frac{2L}{\lambda_1 m (S+1)^2} \|x_0 - x^\star\|^2 + \frac{8\lambda_1(\sigma^2 + M^2)(S+1)}{L},
$$

where $(a)$ follows from $\lambda_1 \leq \frac{2}{3}$ and $(b)$ holds because $0 \leq x \mapsto (x+2)^2$ is non-decreasing and thus

$$\sum_{s=0}^{S-1} (s+2)^2 \leq \int_0^S (x+2)^2 dx \leq \frac{(S+2)^3}{3} \leq \frac{8(S+1)^3}{3}.$$

Denoting

$$\lambda_1^\star \triangleq \frac{L \|x_0 - x^\star\|}{2\sqrt{m}\sqrt{\sigma^2 + M^2}(S+1)^{\frac{3}{2}}},$$

and based on the choice of $\lambda_1 = \min\left\{\frac{2}{3}, \lambda_1^*\right\}$, if $\lambda_1^* \leq \frac{2}{3}$, we have

$$\mathbb{E}\left[F(\tilde{x}_S)\right] - F(x^\star) \leq \frac{8 \|x_0 - x^\star\| \sqrt{\sigma^2 + M^2}}{m^{\frac{1}{2}}(S+1)^{\frac{1}{2}}}.$$

If $\lambda_1^* > \frac{2}{3}$,

$$\mathbb{E}\left[F(\tilde{x}_S)\right] - F(x^\star) \leq \frac{3L \|x_0 - x^\star\|^2}{m(S+1)^2} + \frac{4 \|x_0 - x^\star\| \sqrt{\sigma^2 + M^2}}{m^{\frac{1}{2}}(S+1)^{\frac{1}{2}}}.$$

Thus, we conclude that

$$\mathbb{E}\left[F(\tilde{x}_S)\right] - F(x^\star) \leq \frac{3L \|x_0 - x^\star\|^2}{m(S+1)^2} + \frac{8 \|x_0 - x^\star\| \sqrt{\sigma^2 + M^2}}{m^{\frac{1}{2}}(S+1)^{\frac{1}{2}}}.$$

Substituting $S = K/m$ completes the proof.

### B.2.3 PROOF OF THEOREM 1B

In order to prove Theorem 1b, we need the following known result for the martingale difference (cf. Lemma 2 in Lan et al. (2012)):

**Lemma 2.** *With $N > 0$, let $\xi_0, \xi_1, \ldots, \xi_{N-1}$ be a sequence of i.i.d. random variables, for $t = 0, \ldots, N-1$, $\sigma_t > 0$ be a deterministic number and $\psi_t = \psi_t(\xi_0, \ldots, \xi_t)$ be a deterministic measurable function such that $\mathbb{E}_{\xi_t}[\psi_t] = 0$ a.s. and $\mathbb{E}_{\xi_t}\left[\exp\{\psi_t^2/\sigma_t^2\}\right] \leq \exp\{1\}$ a.s.. Then for any $\Lambda \geq 0$,*

$$\mathrm{Prob}\left\{\sum_{t=0}^{N-1} \psi_t \geq \Lambda \sqrt{\sum_{t=0}^{N-1} \sigma_t^2}\right\} \leq \exp\{-\Lambda^2/3\}.$$

To start with, using Lemma 1 with the parameter mapping (12), we have

$$\frac{1}{1-\beta_s}\left(F(x_{k+1}) - F(x^\star)\right) + \frac{1}{2\alpha_s} \|z_{k+1} - x^\star\|^2$$

$$\leq \frac{\beta_s}{1-\beta_s}\left(F(\tilde{x}_s) - F(x^\star)\right) + \frac{1}{2\alpha_s} \|z_k - x^\star\|^2$$

$$+ \frac{(\|\nabla f(x_k) - \nabla f_{i_k}(x_k)\| + M)^2}{2(\alpha_s^{-1} - L(1-\beta_s))} + \langle \nabla f(x_k) - \nabla f_{i_k}(x_k), z_k - x^\star \rangle$$

$$\leq \frac{\beta_s}{1-\beta_s}\left(F(\tilde{x}_s) - F(x^\star)\right) + \frac{1}{2\alpha_s} \|z_k - x^\star\|^2 + \frac{M^2}{\alpha_s^{-1} - L(1-\beta_s)}$$

$$+ \frac{\|\nabla f(x_k) - \nabla f_{i_k}(x_k)\|^2}{\alpha_s^{-1} - L(1-\beta_s)} + \langle \nabla f(x_k) - \nabla f_{i_k}(x_k), z_k - x^\star \rangle.$$

Summing the above inequality from $k = sm, \ldots, sm + m - 1$ and using the choice $\alpha_s = \frac{\lambda_1}{L(1-\beta_s)}$ with $\lambda_1 \leq \frac{2}{3}$, we obtain

$$\frac{\alpha_s}{1 - \beta_s} \big( F(\tilde{x}_{s+1}) - F(x^\star) \big) + \frac{1}{2m} \left\| z_{(s+1)m} - x^\star \right\|^2$$

$$\leq \frac{\alpha_s \beta_s}{1 - \beta_s} \big( F(\tilde{x}_s) - F(x^\star) \big) + \frac{1}{2m} \left\| z_{sm} - x^\star \right\|^2 + 3\alpha_s^2 M^2$$

$$+ \frac{3\alpha_s^2}{m} \sum_{k=sm}^{sm+m-1} \left\| \nabla f(x_k) - \nabla f_{i_k}(x_k) \right\|^2 + \frac{\alpha_s}{m} \sum_{k=sm}^{sm+m-1} \langle \nabla f(x_k) - \nabla f_{i_k}(x_k), z_k - x^\star \rangle.$$

With our parameter choices, the relations in (14) hold and thus we can telescope the above inequality from $s = S - 1, \ldots, 0$,

$$\frac{\alpha_{S-1}}{1 - \beta_{S-1}} \big( F(\tilde{x}_S) - F(x^\star) \big) \leq \frac{1}{2m} \left\| x_0 - x^\star \right\|^2 + 3M^2 \sum_{s=0}^{S-1} \alpha_s^2$$

$$+ \underbrace{\frac{3}{m} \sum_{k=0}^{K-1} \alpha_{\lfloor k/m \rfloor}^2 \left\| \nabla f(x_k) - \nabla f_{i_k}(x_k) \right\|^2}_{R_4} \tag{15}$$

$$+ \underbrace{\frac{1}{m} \sum_{k=0}^{K-1} \alpha_{\lfloor k/m \rfloor} \langle \nabla f(x_k) - \nabla f_{i_k}(x_k), z_k - x^\star \rangle}_{R_5}.$$

Denoting $\mathcal{V}_k^2 \triangleq \left\| \nabla f(x_k) - \nabla f_{i_k}(x_k) \right\|^2$, $\bar{\alpha} = \sum_{k=0}^{K-1} \alpha_{\lfloor k/m \rfloor}^2 = m \sum_{s=0}^{S-1} \alpha_s^2$, for $R_4$, by Jensen's inequality, we have

$$\mathbb{E} \left[ \exp \left\{ \frac{1}{\bar{\alpha}} \sum_{k=0}^{K-1} \alpha_{\lfloor k/m \rfloor}^2 \mathcal{V}_k^2 / \sigma^2 \right\} \right] \leq \frac{1}{\bar{\alpha}} \sum_{k=0}^{K-1} \alpha_{\lfloor k/m \rfloor}^2 \mathbb{E} \left[ \exp \left\{ \mathcal{V}_k^2 / \sigma^2 \right\} \right] \overset{(\star)}{\leq} \exp\{1\},$$

where $(\star)$ uses the additional assumption $\mathbb{E}_{i_k} \left[ \exp \left\{ \mathcal{V}_k^2 / \sigma^2 \right\} \right] \leq \exp\{1\}$.

Then, based on Markov's inequality, we have for any $\Lambda \geq 0$,

$$\text{Prob} \left\{ \exp \left\{ \frac{1}{\bar{\alpha}} \sum_{k=0}^{K-1} \alpha_{\lfloor k/m \rfloor}^2 \mathcal{V}_k^2 / \sigma^2 \right\} \geq \exp\{\Lambda + 1\} \right\} \leq \exp\{-\Lambda\},$$

$$\text{Prob} \left\{ R_4 \geq (\Lambda + 1)\sigma^2 m \sum_{s=0}^{S-1} \alpha_s^2 \right\} \leq \exp\{-\Lambda\}. \tag{16}$$

For $R_5$, since we have $\mathbb{E}_{i_k} \left[ \alpha_{\lfloor k/m \rfloor} \langle \nabla f(x_k) - \nabla f_{i_k}(x_k), z_k - x^\star \rangle \right] = 0$ and

$$\mathbb{E}_{i_k} \left[ \exp \left\{ \frac{\alpha_{\lfloor k/m \rfloor}^2 \langle \nabla f(x_k) - \nabla f_{i_k}(x_k), z_k - x^\star \rangle^2}{\alpha_{\lfloor k/m \rfloor}^2 \sigma^2 D_X^2} \right\} \right] \leq \mathbb{E}_{i_k} \left[ \exp \left\{ \mathcal{V}_k^2 / \sigma^2 \right\} \right] \leq \exp\{1\},$$

which is based on the "light tail" assumption, using Lemma 2, we obtain

$$\text{Prob} \left\{ R_5 \geq \Lambda \sigma D_X \sqrt{m \sum_{s=0}^{S-1} \alpha_s^2} \right\} \leq \exp\{-\Lambda^2/3\}. \tag{17}$$

Combining (15), (16) and (17), based on the parameter setting (cf. (5)) and using the notation

$$\mathcal{K}_0(m) \triangleq \frac{3Lm \left\| x_0 - x^\star \right\|^2}{(K + m)^2} + \frac{8 \left\| x_0 - x^\star \right\| \sqrt{\sigma^2 + M^2}}{\sqrt{K + m}},$$

$$R_6 \triangleq \frac{12L\sigma^2}{\lambda_1 (S+1)^2} \sum_{s=0}^{S-1} \alpha_s^2 + \frac{4L\sigma D_X}{\lambda_1 (S+1)^2 \sqrt{m}} \sqrt{\sum_{s=0}^{S-1} \alpha_s^2},$$

we conclude that

$$\text{Prob}\left\{F(\tilde{x}_S) - F(x^\star) \le \mathcal{K}_0(m) + \Lambda R_6\right\} \ge 1 - (\exp\{-\Lambda^2/3\} + \exp\{-\Lambda\}).$$

For $R_6$, using the choice of $\alpha_s$ and $\lambda_1$, we obtain

$$R_6 \le \frac{4\sqrt{6}\sigma D_X}{3\sqrt{K+m}} + \frac{8\lambda_1\sigma^2(S+1)}{L} \le \frac{4\sqrt{6}\sigma D_X}{3\sqrt{K+m}} + \frac{4\sigma^2 \|x_0 - x^\star\|}{\sqrt{K+m}\sqrt{\sigma^2 + M^2}}$$

$$\le \frac{4\sigma\left(3\|x_0 - x^\star\| + \sqrt{6}D_X\right)}{3\sqrt{K+m}},$$

which completes the proof.

### B.2.4 PROOF OF THEOREM 2

Using Assumption (c), Lemma 1 with

$$\begin{cases} x = x_k^{j_k} \\ z = z_k \\ z^+ = z_{k+1} \\ y = \phi_{j_k}^k \\ y^+ = \phi_{j_k}^{k+1} \\ \alpha = \alpha_k \\ \beta = \beta_k \end{cases},$$

and taking expectation, if $\alpha_k(1 - \beta_k) < 1/L$, we have

$$\frac{1}{1-\beta_k}\mathbb{E}_{i_k,j_k}\left[F(\phi_{j_k}^{k+1}) - F(x^\star)\right] + \frac{1}{2\alpha_k}\mathbb{E}_{i_k,j_k}\left[\|z_{k+1} - x^\star\|^2\right]$$

$$\le \frac{\beta_k}{1-\beta_k}\mathbb{E}_{j_k}\left[F(\phi_{j_k}^k) - F(x^\star)\right] + \frac{1}{2\alpha_k}\|z_k - x^\star\|^2 + \frac{(\sigma + M)^2}{2(\alpha_k^{-1} - L(1-\beta_k))}. \tag{18}$$

Note that

$$\mathbb{E}_{i_k,j_k}\left[F(\phi_{j_k}^{k+1}) - F(x^\star)\right]$$

$$= \mathbb{E}_{i_k,j_k}\left[\sum_{j=1}^m \left(F(\phi_j^{k+1}) - F(x^\star)\right) - \sum_{j\ne j_k}^m \left(F(\phi_j^k) - F(x^\star)\right)\right]$$

$$= \mathbb{E}_{i_k,j_k}\left[\sum_{j=1}^m \left(F(\phi_j^{k+1}) - F(x^\star)\right)\right] - \mathbb{E}_{j_k}\left[\sum_{j\ne j_k}^m \left(F(\phi_j^k) - F(x^\star)\right)\right].$$

Dividing both sides of (18) by $m$ and then adding $\frac{1}{(1-\beta_k)m}\mathbb{E}_{j_k}\left[\sum_{j\ne j_k}^m \left(F(\phi_j^k) - F(x^\star)\right)\right]$ to both sides, we obtain

$$\frac{1}{1-\beta_k}\mathbb{E}_{i_k,j_k}\left[\frac{1}{m}\sum_{j=1}^m F(\phi_j^{k+1}) - F(x^\star)\right] + \frac{1}{2\alpha_k m}\mathbb{E}_{i_k,j_k}\left[\|z_{k+1} - x^\star\|^2\right]$$

$$\le -\frac{1}{m}\mathbb{E}_{j_k}\left[F(\phi_{j_k}^k) - F(x^\star)\right] + \frac{1}{1-\beta_k}\left(\frac{1}{m}\sum_{j=1}^m F(\phi_j^k) - F(x^\star)\right) + \frac{1}{2\alpha_k m}\|z_k - x^\star\|^2$$

$$+ \frac{(\sigma + M)^2}{2m(\alpha_k^{-1} - L(1-\beta_k))}$$

$$= \frac{1 - \frac{1-\beta_k}{m}}{1-\beta_k}\left(\frac{1}{m}\sum_{j=1}^m F(\phi_j^k) - F(x^\star)\right) + \frac{1}{2\alpha_k m}\|z_k - x^\star\|^2 + \frac{(\sigma + M)^2}{2m(\alpha_k^{-1} - L(1-\beta_k))}. \tag{19}$$

It can be verified that with our parameters choice: $\beta_k = \frac{k/m}{k/m+2}$ and $\alpha_k = \frac{\lambda_2}{L(1-\beta_k)}$, the following holds for $k \geq 0$,

$$\alpha_{k+1} \frac{1 - \frac{1-\beta_{k+1}}{m}}{1 - \beta_{k+1}} \leq \frac{\alpha_k}{1 - \beta_k} \text{ and } \beta_0 = 0.$$

Note that since our analysis aims at providing intuition, we do not refine the choice of $\alpha_s$ as in (Hu et al., 2009; Ghadimi & Lan, 2012). Then, we can telescope (19) from $k = K - 1, \ldots, 0$, which results in

$$\frac{\alpha_{K-1}}{1 - \beta_{K-1}} \mathbb{E}\left[ \frac{1}{m} \sum_{j=1}^{m} F(\phi_j^K) - F(x^\star) \right] + \frac{1}{2m} \mathbb{E}\left[ \|z_K - x^\star\|^2 \right]$$

$$\leq \frac{\lambda_2(m-1)}{Lm} \left( F(x_0) - F(x^\star) \right) + \frac{1}{2m} \|x_0 - x^\star\|^2 + \sum_{k=0}^{K-1} \frac{\alpha_k(\sigma + M)^2}{2m(\alpha_k^{-1} - L(1 - \beta_k))}.$$

Using the definition of $\bar{\phi}^K$ and convexity, we obtain

$$\mathbb{E}\left[ F(\bar{\phi}^K) - F(x^\star) \right] \leq \frac{1 - \beta_{K-1}}{\alpha_{K-1}} \left( \frac{\lambda_2(m-1)}{Lm} \left( F(x_0) - F(x^\star) \right) + \frac{1}{2m} \|x_0 - x^\star\|^2 \right)$$

$$+ \frac{1 - \beta_{K-1}}{\alpha_{K-1}} \sum_{k=0}^{K-1} \frac{\alpha_k(\sigma + M)^2}{2m(\alpha_k^{-1} - L(1 - \beta_k))}$$

$$\overset{(a)}{=} \frac{4(m-1)\left( F(x_0) - F(x^\star) \right)}{m \left( \frac{K-1}{m} + 2 \right)^2} + \frac{2L \|x_0 - x^\star\|^2}{\lambda_2 m \left( \frac{K-1}{m} + 2 \right)^2}$$

$$+ \frac{3\lambda_2(\sigma + M)^2}{2Lm \left( \frac{K-1}{m} + 2 \right)^2} \sum_{k=0}^{K-1} \left( \frac{k}{m} + 2 \right)^2$$

$$\overset{(b)}{\leq} \frac{4(m-1)\left( F(x_0) - F(x^\star) \right)}{m \left( \frac{K-1}{m} + 2 \right)^2} + \frac{2L \|x_0 - x^\star\|^2}{\lambda_2 m \left( \frac{K-1}{m} + 2 \right)^2} \qquad (20)$$

$$+ \frac{4\lambda_2(\sigma + M)^2 \left( \frac{K-1}{m} + 2 \right)}{L},$$

where $(a)$ uses $\lambda_2 \leq \frac{2}{3}$, $(b)$ follows from simple integration arguments and that $\frac{K}{m} + 2 \leq 2 \left( \frac{K-1}{m} + 2 \right)$ since $K \geq 1, m \geq 1$.

Based on the choice of

$$\lambda_2 = \min \left\{ \frac{2}{3}, \frac{L \|x_0 - x^\star\|}{\sqrt{2m}(\sigma + M) \left( \frac{K-1}{m} + 2 \right)^{\frac{3}{2}}} \right\},$$

(20) can be further upper bounded as

$$\mathbb{E}\left[ F(\bar{\phi}^K) - F(x^\star) \right] \leq \frac{4(m-1)\left( F(x_0) - F(x^\star) \right)}{m \left( \frac{K-1}{m} + 2 \right)^2} + \frac{3L \|x_0 - x^\star\|^2}{m \left( \frac{K-1}{m} + 2 \right)^2} + \frac{4\sqrt{2} \|x_0 - x^\star\| (\sigma + M)}{m^{\frac{1}{2}} \left( \frac{K-1}{m} + 2 \right)^{\frac{1}{2}}}.$$

### B.3 CONNECTIONS BETWEEN AM1-SGD AND KATYUSHA

The discussion in this section aims to shed light on the understanding of the experimental results, which also shows some interesting relations between AM1-SGD and Katyusha.

The high level idea of Katyusha momentum is that it works as a "magnet" inside an epoch of SVRG updates, which "stabilizes" the iterates so as to make Nesterov's momentum effective (Allen-Zhu, 2018). In theory, the key effect of Katyusha momentum is that it allows the tightest possible variance bound for the stochastic gradient estimator of SVRG (cf. Lemma 2.4 and its comments in Allen-Zhu (2018)). In this sense, we can interpret Katyusha momentum as a variance reducer that further reduces the variance of SVRG. Below we show the similarity between the construction of Katyusha and AM1-SGD, based on which we conjecture that the amortized momentum can also reduce the variance of SGD (and thus increase the robustness). However, in theory, following a similar analysis of Katyusha, we cannot guarantee a reduction of $\sigma$ in the worst case.

**Deriving AM1-SGD from Katyusha** Katyusha has the following scheme (non-proximal, in the original notations, $\sigma$ is the strong convexity parameter, cf. Algorithm 1 with Option I in Allen-Zhu (2018))[12]:

> **Initialize:** $\widetilde{x}^0 = y_0 = z_0 = x_0, \eta = \frac{1}{3L}, \omega = 1 + \alpha\sigma$.
> 1: **for** $s = 0, \ldots, S - 1$ **do**
> 2:     Compute and store $\nabla f(\widetilde{x}^s)$.
> 3:     **for** $j = 0, \ldots, m - 1$ **do**
> 4:         $k = sm + j$.
> 5:         $x_k = \tau_1 \cdot z_k + \tau_2 \cdot \widetilde{x}^s + (1 - \tau_1 - \tau_2) \cdot y_k$.
> 6:         $\widetilde{\nabla}_k = \nabla f_{i_k}(x_k) - \nabla f_{i_k}(\widetilde{x}^s) + \nabla f(\widetilde{x}^s)$.
> 7:         $z_{k+1} = z_k - \alpha \cdot \widetilde{\nabla}_k$.
> 8:         $y_{k+1} = x_k - \eta \cdot \widetilde{\nabla}_k$.
> 9:     **end for**
> 10:   $\widetilde{x}^{s+1} = \left(\sum_{j=0}^{m-1} \omega^j\right)^{-1} \cdot \sum_{j=0}^{m-1} \omega^j \cdot y_{sm+j+1}$.
> 11: **end for**
> **Output:** $\widetilde{x}^S$.

We can eliminate the sequence $\{z_k\}$ in this scheme. Note that in the parameter setting of Katyusha, we have $\eta = \alpha\tau_1$, and thus

$$
\begin{aligned}
x_{k+1} &= \tau_1 \cdot z_{k+1} + \tau_2 \cdot \widetilde{x}^s + (1 - \tau_1 - \tau_2) \cdot y_{k+1} \\
&= \tau_1 \cdot z_k - \eta \cdot \widetilde{\nabla}_k + \tau_2 \cdot \widetilde{x}^s + (1 - \tau_1 - \tau_2) \cdot y_k + (1 - \tau_1 - \tau_2) \cdot (y_{k+1} - y_k) \\
&= x_k - \eta \cdot \widetilde{\nabla}_k + (1 - \tau_1 - \tau_2) \cdot (y_{k+1} - y_k) \\
&= y_{k+1} + (1 - \tau_1 - \tau_2) \cdot (y_{k+1} - y_k).
\end{aligned}
$$

Hence, the inner loops can be written as

$$
\begin{aligned}
y_{k+1} &= x_k - \eta \cdot \widetilde{\nabla}_k, \\
x_{k+1} &= y_{k+1} + (1 - \tau_1 - \tau_2) \cdot (y_{k+1} - y_k),
\end{aligned}
$$

which is the Nesterov's scheme (scheme (1)). At the end of each inner loop (when $k = sm + m - 1$),

$$
x_{(s+1)m} = \tau_1 \cdot z_{(s+1)m} + \tau_2 \cdot \widetilde{x}^s + (1 - \tau_1 - \tau_2) \cdot y_{(s+1)m},
$$

while at the beginning of the next inner loop,

$$
x_{(s+1)m} = \tau_1 \cdot z_{(s+1)m} + \tau_2 \cdot \widetilde{x}^{s+1} + (1 - \tau_1 - \tau_2) \cdot y_{(s+1)m},
$$

which means that we need to set $x_{(s+1)m} \leftarrow x_{(s+1)m} + \tau_2 \cdot (\widetilde{x}^{s+1} - \widetilde{x}^s)$ (reassign the value of $x_{(s+1)m}$).

Then, the following is an equivalent scheme of Katyusha:

---

[12] We change the notation $x_{k+1}$ to $x_k$.

**Initialize:** $\widetilde{x}^0 = y_0 = x_0, \eta = \frac{1}{3L}, \omega = 1 + \alpha\sigma$.
1: **for** $s = 0, \ldots, S - 1$ **do**
2:    **for** $j = 0, \ldots, m - 1$ **do**
3:       $k = sm + j$.
4:       $y_{k+1} = x_k - \eta \cdot \widetilde{\nabla}_k$.
5:       $x_{k+1} = y_{k+1} + (1 - \tau_1 - \tau_2) \cdot (y_{k+1} - y_k)$.
6:    **end for**
7:    $\widetilde{x}^{s+1} = \left( \sum_{j=0}^{m-1} \omega^j \right)^{-1} \cdot \sum_{j=0}^{m-1} \omega^j \cdot y_{sm+j+1}$.
8:    $x_{(s+1)m} \leftarrow x_{(s+1)m} + \tau_2 \cdot (\widetilde{x}^{s+1} - \widetilde{x}^s)$.
9: **end for**
**Output:** $\widetilde{x}_S$.

Now it is clear that the inner loops use Nesterov's momentum and the Katyusha momentum is injected for every $m$ iterations. If we replace the SVRG estimator $\widetilde{\nabla}_k$ with $\nabla f_{i_k}(x_k)$, set $1 - \tau_1 - \tau_2 = 0$, which is to eliminate Nesterov's momentum, and use a uniform average for $\widetilde{x}^{s+1}$, the above scheme becomes exactly AM1-SGD (Algorithm 1).

If we only replace the SVRG estimator $\widetilde{\nabla}_k$, the scheme can be regarded as adding amortized momentum to M-SGD. This scheme requires tuning the ratio of Nesterov's momentum and amortized momentum. In our preliminary experiments, after suitable tuning, we observed some performance improvement. However, this scheme increases the complexity, which we do not consider it worthwhile.

A recent work (Zhou et al., 2018) shows that when $1 - \tau_1 - \tau_2 = 0$, which is to solely use Katyusha momentum, one can still derive optimal rates and the algorithm is greatly simplified. Their proposed algorithm (i.e., MiG) is structurally more similar to AM1-SGD.

## C  MISCELLANIES

### C.1  COMPARISON OF SGD AND M-SGD

Ma & Yarats (2019) normalize the momentum buffer of M-SGD, which results in the following formulation ($\alpha \in \mathbb{R}, v^{qh} \in \mathbb{R}^d, v_0^{qh} = \mathbf{0}$):

$$v_{k+1}^{qh} = \beta \cdot v_k^{qh} + (1 - \beta) \cdot \nabla f_{i_k}(x_k),$$
$$x_{k+1} = x_k - \alpha \cdot (\beta \cdot v_{k+1}^{qh} + (1 - \beta) \cdot \nabla f_{i_k}(x_k)), \text{ for } k \geq 0.$$

This scheme is equivalent to the PyTorch formulation (scheme (3)) through $v_k^{pt} = (1 - \beta)^{-1} \cdot v_k^{qh}$ and $\eta = \alpha(1 - \beta)$.

Based on this formulation, $\alpha$ is understood as the effective learning rate (i.e., the vector it scales has the same cardinality as a gradient) and the experiments in Ma & Yarats (2019) were conducted with fixed $\alpha = 1$. Their results indicate that when using the same effective learning rate, M-SGD and SGD achieve similar performance and thus they suspect that the benefit of momentum basically comes from using sensible learning rates.

Here we provide some intuition on their results based on convex analysis. For simplicity, we consider deterministic smooth convex optimization. In theory, to obtain the optimal convergence rate, the effective learning rate $\alpha$ is set to a very large $O(k/L)$, which can be derived from Theorem 1 or Theorem 2 by setting $\sigma = 0, M = 0, m = 1$ (then $\lambda_1$ or $\lambda_2$ is always $\frac{2}{3}$ since the other term is $\infty$). If we fix $\alpha = \frac{2}{3L}$ for both methods, GD has an $O(1/K)$ convergence rate (cf. Theorem 2.1.13 in Nesterov (2013b)). For the Nesterov's method, if we use $\beta_k = \frac{k}{k+2}$, it has the convergence rate (applying Lemma 1):

$$\frac{1}{1 - \beta_k}\big(F(y_{k+1}) - F(x^\star)\big) + \frac{3L}{4}\|z_{k+1} - x^\star\|^2 \leq \frac{\beta_k}{1 - \beta_k}\big(F(y_k) - F(x^\star)\big) + \frac{3L}{4}\|z_k - x^\star\|^2,$$

$$F(y_K) - F(x^\star) \leq \frac{3L\|x_0 - x^\star\|^2}{2(K + 1)}.$$

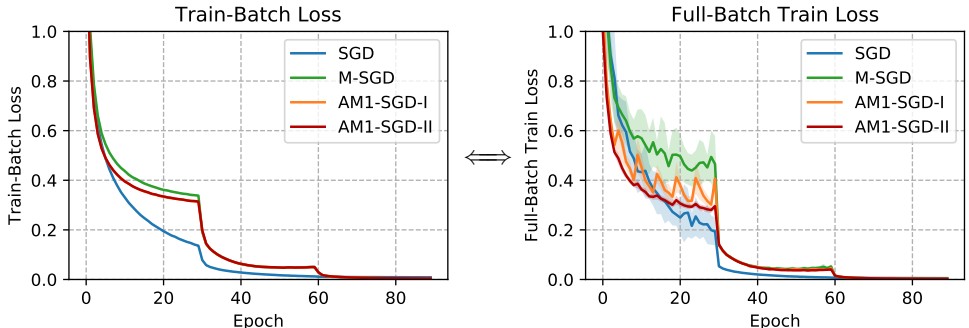

Figure 15: Train-batch loss vs. full-batch loss. Best viewed in color.

Thus, in this case, both GD and the Nesterov's method yield an $O(1/K)$ rate, and thus we expect them to have similar performance. This analysis suggests that the acceleration effect basically comes from choosing a large effective learning rate, which corresponds to the observations in Ma & Yarats (2019).

However, what is special about the Nesterov's method is that it finds a legal way to adopt a large $\alpha$ that breaks the $1/L$ limitation. If GD uses the same large $\alpha$, we would expect it to be unstable and potentially diverge. In this sense, Nesterov's momentum can be understood as a "stabilizer". In our basic case study (ResNet34 on CIFAR-10), if we align the effective learning rate and set $\gamma = 1.0$ for SGD, the final accuracy is improved but the performance is highly unstable and not robust, which is $2.205\%$ average STD of test accuracy over 5 runs. The significance of QHM (Ma & Yarats, 2019) is that with suitable tuning, it achieves much faster convergence without changing the effective learning rate. Our work uses the convergence behavior of SGD as a reference to reflect and to understand the features of our proposed momentum, which is why we set $\gamma = \eta$.

## C.2  TRAINING EVALUATION

Due to the mechanism of back-propagation, evaluating train-batch loss basically incurs no overhead. It can be efficiently used to indicate the training progress. However, it can only be treated as a coarse approximation to the full-batch loss as shown in Figure 1c. If batch normalization (Ioffe & Szegedy, 2015) or dropout (Srivastava et al., 2014) is used in training, the model changes during the training phase, which makes train-batch loss less accurate. More importantly, train-batch loss is always observed to be statistically stable, which omits many important characteristics of an optimizer such as robustness, oscillations, etc. We include a comparison of train-batch loss and full-batch loss on training ResNet18 with pre-activation on CIFAR-10 (the experiment in Section 4) in Figure 15.

We also notice that for different optimizers, even if their convergences on train-batch loss are indistinguishable, their convergences on test accuracy can vary greatly. We show two examples in Figure 16, where the ImageNet experiment is from Section 5 and the CIFAR-10 experiment is from Table 4 with $m = 10$.

## D  EXPERIMENTAL SETUP

All of our experiments were conducted using PyTorch (Paszke et al., 2017) library.

### D.1  CLASSIFICATION SETUP

**CIFAR-10 & CIFAR-100**  Our implementation (e.g., ResNet and DenseNet implementations, data pre-processing) generally follows the repository https://github.com/kuangliu/pytorch-cifar. All the CIFAR experiments in this paper used a single GPU in a mix of RTX2080Ti, TITAN Xp and TITAN V. The batch size is fixed to 128. We used cross-entropy loss with 0.0005 weight decay and used batch normalization (Ioffe & Szegedy, 2015). Data augmentation includes random 32-pixel crops with a padding of 4-pixel and random horizontal flips with

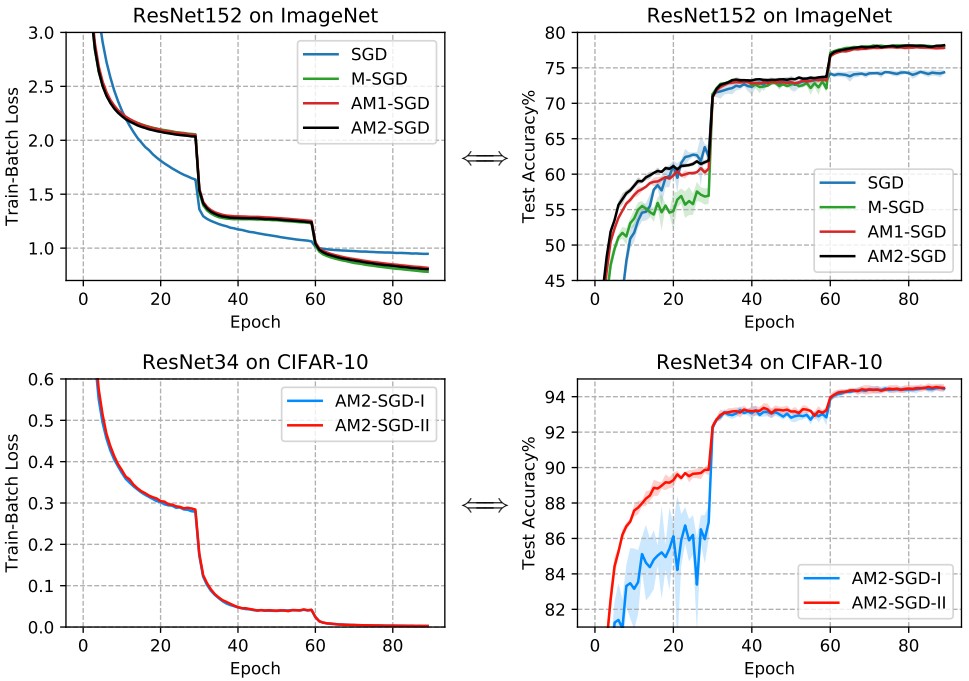

Figure 16: Train-batch loss vs. test accuracy. Best viewed in color.

0.5 probability. We used step (or multi-step) learning rate scheduler with a decay factor 10. For the CIFAR-10 experiments, we trained 90 epochs and decayed the learning rate every 30 epochs. For the CIFAR-100 experiments, we trained 300 epochs and decayed the learning rate at 150 epoch and 225 epoch following the settings in DenseNet (Huang et al., 2017).

**ImageNet** In the ImageNet experiments, we tried both ResNet50 and ResNet152 (He et al., 2016b). The training strategy is the same as the PyTorch's official repository `https://github.com/pytorch/examples/tree/master/imagenet`, which uses a batch size of 256. The learning rate starts at 0.1 and decays by a factor of 10 every 30 epochs. Also, we applied weight decay with 0.0001 decay rate to the model during the training. For the data augmentation, we applied random 224-pixel crops and random horizontal flips with 0.5 probability. Here, we run all experiments across 8 NVIDIA P100 GPUs for 90 epochs.

### D.2 LANGUAGE MODEL SETUP

We followed the implementation in the repository `https://github.com/salesforce/awd-lstm-lm` and trained word level Penn Treebank with LSTM without fine-tuning or continuous cache pointer augmentation for 750 epochs. The experiments were conducted on a single RTX2080Ti. We used the default hyper-parameter tuning except for learning rate and momentum: The LSTM has 3 layers containing 1150 hidden units each, embedding size is 400, gradient clipping has a maximum norm 0.25, batch size is 80, using variable sequence length, dropout for the layers has probability 0.4, dropout for the RNN layers has probability 0.3, dropout for the input embedding layer has probability 0.65, dropout to remove words from embedding layer has probability 0.1, weight drop (Merity et al., 2017) has probability 0.5, the amount of $\ell 2$-regularization on the RNN activation is 2.0, the amount of slowness regularization applied on the RNN activation is 1.0 and all weights receive a weight decay of 0.0000012.

