# OpenReview forum: "Amortized Nesterov's Momentum: Robust and Lightweight  Momentum for Deep Learning"
_ICLR.cc/2020/Conference — Reject_

### Official Review · AnonReviewer3 · 2019-10-21
**Official Blind Review #3**

**Rating:** 8

**Review:**

This paper provides a new simple method to incorporate Nesterov momentum into standard SGD for deep learning, with good empirical and theoretical results. Overall I think this paper should be accepted, some minor comments follow.

At no point does Polyak's heavy ball method get mentioned, even though the variant of Nesterov acceleration you are considering is very similar to it (since the momentum parameter is fixed, which is not the usual form of Nesterov except in the strongly convex case). It would be beneficial to delineate how this is or isn't related to heavy ball.

The experiments would benefit from a wall-clock time comparison too, rather than just epochs since these new methods would be slower (but presumably not by much).

The appendix is huge with most of the technical details relegated there which I did not read fully. I think this impacts the readability significantly, though not grounds for rejection. Perhaps it suggests that a conference with a small page limit is not the best venue?

It seems that SGD still has better convergence early on. The authors suggest their method fixes this (relative to standard nesterov SGD) but it doesn't seem to be quite as good as SGD. Can you explain or discuss why this is still the case?

The assumptions require some explanation, they are just listed with no context. What are they and why are they useful?

Step size "should be constrained to O(1/L)" is misleading, you should say explicitly that step-size <= 1/L (or whatever it is depending on the algorithm).

Some of the writing is a bit strange / sloppy, e.g.:
"AM2-SGD is a bit tricky in randomness"
"However, full-batch loss is way too expensive to evaluate."

In Algorithm 1 AM1-SGD:
"xk+1 ← xk+1 + β · (˜x+ − x˜)"
doesn't parse because x_{k+1} appears twice.

Missing references:

Accelerated proximal algorithms:

*) Beck and Teboulle: A Fast Iterative Shrinkage-Thresholding Algorithm
for Linear Inverse Problems

*) Nesterov: Gradient Methods for Minimizing Composite Objective Function,

Restarting (slightly different to your approach but still relevant):

*) O'Donoghue: Adaptive Restart for Accelerated Gradient Schemes

**Experience Assessment:**

I have published in this field for several years.

**Review Assessment: Checking Correctness Of Derivations And Theory:**

I did not assess the derivations or theory.

**Review Assessment: Checking Correctness Of Experiments:**

I assessed the sensibility of the experiments.

**Review Assessment: Thoroughness In Paper Reading:**

I read the paper at least twice and used my best judgement in assessing the paper.

---

> ### Author Response · Authors · 2019-11-11
> **Response to Reviewer 3**
>
> Thank you for your positive feedbacks.
>
> We have added more explanations to the assumptions, changed $O(1/L)$ to strict upper bounds, improved the writing and the presentation of Algorithm 1, made the main text more independent from the appendices, and cited missing references in the revision.
>
> [Compare with Polyak’s heavy ball (HB)]
> If $m=1$, AM1-SGD reduces to M-SGD and its difference from HB has been carefully studied in [2].
>
> Here we consider a simple case $m=2$. We only compare 2 iterations $x_0\rightarrow x_1\rightarrow x_2$ of AM1-SGD (Option I) and HB. $g_0$ and $g_1$ denote the gradients evaluated at $x_0$ and $x_1$ and we mark the learning rate and momentum parameter ($>0$) for AM1-SGD and HB as $(\eta, \beta)$ and $(\alpha, \gamma)$, respectively.
>
> Since the first iteration of both AM1-SGD and HB is SGD, if $\alpha = \eta$, they generate the same $x_1$ (and thus the same $g_1$).
>
> After the second iteration, AM1-SGD produces
> $$
> x_2 = x_0 - \eta(1 + \beta)\cdot g_0 - \eta(1 + \beta/2)\cdot g_1,
> $$
> and HB produces
> $$
> x_2 = x_0 - \alpha(1 + \gamma)\cdot g_0 - \alpha \cdot g_1.
> $$
> It is unlikely for them to produce the same $x_2$.
>
>
> [Wall-clock time]
> We recorded all the wall-clock time in the CIFAR experiments. However, we observed that even on the same type of GPUs, the running times fluctuate a lot and do not exhibit a clear trend with increasing $m$ (For AM1-SGD, a larger $m$ leads to a lower amortized cost). Thus, we did not report them in the paper. The running time is improved by about 2%-3% for AM1-SGD ($m=5$) compared with M-SGD (measured on the same GPU and using the same random batches). AM2-SGD ($m=5$) is slightly slower than M-SGD.
>
> [SGD still has better convergence early on]
> Here we provide some possible explanations in the (strongly) convex setting (The objective is $L$-smooth and $\mu$-strongly convex, we denote $\kappa = L/\mu$ as the condition number):
> In theory, accelerated algorithms (e.g., NAG, Katyusha) are only faster in the ill-conditioned case (when $\kappa$ is very large). When $\kappa$ is small (the well-conditioned case), their convergence rates are the same as non-accelerated algorithms (e.g., GD, SVRG).
> In practice, it is always observed that in the well-conditioned case, non-accelerated algorithms perform better than accelerated algorithms and increasing momentum only deteriorates the performance. In the ill-conditioned case, it is frequently observed that non-accelerated algorithms have a faster early convergence but are surpassed by accelerated algorithms later on, just like the results of SGD and M-SGD given in the paper.
> Our conjecture is that in the early stage of training, the problem is locally well-conditioned and momentum hurts the performance. When the iterate enters an ill-conditioned region, the momentum becomes effective and provides acceleration.
>
> This negative effect is like an intrinsic disadvantage of momentum that is “transferred” to deep learning training. The amortization technique eases this effect by trading acceleration with variance control (using the intuition of Section 4). A larger $m$ can produce a faster early convergence than SGD (Figure 6) but reduces more final performance. Perhaps using an adaptive choice of $m$ is better.
>
>
> [2] Ilya Sutskever, James Martens, George Dahl, Geoffrey Hinton. On the importance of initialization and momentum in deep learning, ICML 2013.

---

### Official Review · AnonReviewer1 · 2019-10-23
**Official Blind Review #1**

**Rating:** 3

**Review:**

%% Post Author Response comments %%
Thank you for your detailed response/revision.

1 - Introducing “m-times” larger momentum: Somehow, this is not a particularly intuitive statement or one that reflects clearly in a theoretical bound. Since we are getting to issues surrounding the use of momentum with stochastic optimization, I would like to make a note that the performance of these algorithms more broadly aren't quite sketched out for their use in broader stochastic optimization. In particular, despite broad use in practice, it is unclear if standard variants of Nesterov acceleration/Heavy Ball method achieve "acceleration" in stochastic optimization. See for e.g., the work of Kidambi et al ICLR 2018 (“On the insufficiency of existing momentum schemes for stochastic optimization”) - where, the argument was that these methods were designed for deterministic optimization (where we get exact gradients) - in fact, that paper empirically as well as theoretically shows that these schemes do not offer acceleration in a precise sense compared to specialized algorithms for stochastic optimization. It is unclear if the proposed algorithms can offer a similar improvement over SGD in a provable sense, even for the specific examples described in their paper.

2 - The point about theory (just as you mention) is that it doesn’t directly apply towards the simulations, nor, do they improve on already known algorithms - so I am unable to see the point that these results present broader implications that can guide practice.

3 - The response doesn’t address the fact that for the theory bounds presented in the paper to hold (even in the convex settings described), one requires knowledge of parameters that are not known a-priori, and are often fairly difficult to estimate. So the performance of the algorithm in practice may quite significantly be away from the bounds described in the paper.

While I appreciate the points and revision made by the authors, I still believe the paper requires some rethinking to present their results (and this includes more detailed comparisons to existing works) in order to make a case towards broader practical applications.


%%. %%

This paper considers robustness issues faced by Nesterov’s Acceleration used with mini-batch stochastic gradients for training Deep Models. In particular, the paper proposes amortized momentum, an algorithm that offers a way to handle these issues. The paper in general is well written and easy to follow.

The paper proposes algorithms AM-SGD1 and AM-SGD2 and presents extensive results regarding their complexity analysis on convex problems and their performance when training neural networks. The algorithms require storing one more model’s worth of storage compared to standard momentum based methods (which can be viewed as a drawback in certain cases).

Comments:

[1] I am concerned about the motivation behind this paper - which, according to the paper is that Nesterov’s accelerated gradient method with stochastic gradients has huge initial fluctuations. The issue with regards to more fluctuations of the initial performance is natural given how aggressive these accelerated methods work. As long as this is not a reason/cause for worse terminal performance (which doesn’t seem to be the case), I am unable to see why large initial fluctuations are concerning.

[2] Theory: The theory bounds for this problem setting do not appear to improve over known bounds in the literature. As a side note, the work of Hu et al. “Accelerated Gradient Methods for Stochastic Optimization and Online Learning” is highly related to this paper’s theoretical aspects, setup and bounds. Furthermore, this bounded variance noise model for stochastic gradients, while being theoretically useful (and important), is often very detached from practice (as this implies that the domain is bounded and we perform projections of iterates whenever they go outside the set - such aspects hardly reflect on practical SGD implementations). Using this as a means to reason about robustness of the proposed algorithm (for e.g. remarks for theorem 1a. and in conclusions) appears to be a big leap that may lead to potentially misleading conclusions.

[3] In order to run the algorithm to achieve the theoretical bounds claimed (in theorems 1 and 2), it appears that the stepsize \alpha_s depends on unknown quantities such as initial distance to opt, noise variance etc.

[4] The claim in page 2 about comparing SGD and M-SGD says that the stepsize in deterministic and stochastic optimization is constrained to be O(1/L) is rather misleading. In realistic practical implementation of SGD with a multiplicative noise oracle, one really has to use a much smaller stepsize than 1/L. This in a sense leads back to point[2] about the unrealistic nature of bounded variance assumptions for understanding SGD based methods used in the context of Machine Learning. They are better suited for understanding stochastic methods in black-box optimization (as opposed to considering Machine Learning problems).

My take is that even if the authors justify novelty in terms of theory results (which, to my knowledge is limited compared to existing literature), rewriting the paper by considering its theoretical merit and presenting empirical results (even as considered in this paper) of this algorithm (without attempting to make very strong connections to explain issues experienced in non-convex training of neural networks, since the theory works in vastly different settings under restrictive assumptions) can be appreciated by appropriate sections of audience (both in theory as well as optimization for deep learning communities).

**Experience Assessment:**

I have published one or two papers in this area.

**Review Assessment: Checking Correctness Of Derivations And Theory:**

I assessed the sensibility of the derivations and theory.

**Review Assessment: Checking Correctness Of Experiments:**

I assessed the sensibility of the experiments.

**Review Assessment: Thoroughness In Paper Reading:**

I read the paper at least twice and used my best judgement in assessing the paper.

---

> ### Author Response · Authors · 2019-11-11
> **Response to Reviewer 1**
>
> Thanks for carefully reading our paper and your thoughtful comments.
>
> [1, Motivation]
> We admit that we should have made the motivation clearer in the introduction. Our motivation is to design a favorable alternative for M-SGD. We have listed all the pros and cons of AM1-SGD in the introduction and emphasized the “easy-to-use” feature in the revision. In the original submission, we tried to point out the robustness of M-SGD, for which the amortization technique has the most obvious effect: the amortized momentum is aggressive (intuitively, AM1-SGD is injecting $m$-times larger momentum (compared with M-SGD) into plain SGD every $m$ iterations), but it is able to reduce the fluctuations in the initial stage (comparing SGD and AM1-SGD with Option I in Figure 3). We agree that the terminal performance of M-SGD is not affected by the initial fluctuations, which makes the motivation unclear in the introduction. Thus, we have removed the related paragraph in the introduction in the revision.
>
> [2, Theory]
> - The theory bounds for this problem setting do not appear to improve.
> The theory bounds are already optimal, and actually the novelty of this paper is about how we modified M-SGD (AC-SA) to fit our high level idea while maintaining the optimality. Our purpose of the theory section is mainly to understand what has changed when amortizing Nesterov’s momentum (the effect of $m$). While these results do not hold for deep learning problems in general, they shed light on the tuning of $m$ in deep learning applications. We have revised the first paragraph in Section 4 to make it clearer.
>
> - The work of Hu et al.
> [Hu et al., 2009] and the work [Ghadimi & Lan, 2012] both focus on extending AC-SA [Lan, 2012] to strongly convex setting and refining its parameter choices. In comparison, the assumptions in [Ghadimi & Lan, 2012] are more general and more reasonable ([Hu et al., 2009] considers the unconstrained case and assumes that an algorithm-generated sequence is bounded). We added [Hu et al., 2009] to the reference in the revision.
>
> - The bounded variance noise model for stochastic gradients.
> We chose the stochastic bounded variance model since we want to understand why the amortization trick can reduce fluctuations. It actually turns out that in the deterministic setting ($\sigma=0$), amortization (increasing $m$) is meaningless (Theorem 1a and 2).
>
> - The domain is bounded and we perform projections of iterates.
> Compactness is assumed only in Theorem 1b. Theorem 1a and 2 hold in unconstrained setting ($X = \mathbb{R}^d, h \equiv 0$).
>
> - Using this as a means to reason … is a big leap.
> We admit that the statements about using the theory to reason the empirical results in the original submission are misleading, and we have revised the statements to say that the theory can provide a better understanding of the empirical results.
>
> [3, $\alpha_s$ depends on unknown quantities]
> We did not follow [Ghadimi & Lan, 2012] and [Hu et al., 2009] to derive a better parameter setting because it does not contribute to the understanding of $m$ but complicates the analysis. We will add some comments about this in the next revision.
>
> [4, $1/L$]
> Thanks for pointing this out. We have modified all the $O(1/L)$ to strict upper bounds.
> About using intuitions from convex analysis: Nesterov’s momentum was migrated from convex optimization to deep learning by [Sutskever et al., 2013] and it turned out to be quite successful. We actually followed [Sutskever et al., 2013] to design tricks in convex analysis and explore their benefits in deep learning applications.
>
> [Hu et al., 2009], Accelerated Gradient Methods for Stochastic Optimization and Online Learning, 2009
> [Ghadimi & Lan, 2012]. Optimal stochastic approximation algorithms for strongly convex stochastic composite optimization i: A generic algorithmic framework, 2012
> [Lan, 2012]. An optimal method for stochastic composite optimization, 2012
> [Sutskever et al., 2013]. On the importance of initialization and momentum in deep learning, 2013

---

### Official Review · AnonReviewer2 · 2019-11-03
**Official Blind Review #2**

**Rating:** 3

**Review:**

The authors proposed Amortized Nesterov’s Momentum, a variant of Nesterov’s momentum that utilizes several past iterates, instead of one iterate, to provide the momentum.  The goal is to have more robust iterates, faster convergence in the early stage and higher efficiency. The authors designed two different realizations, AM1-SGD and AM2-SGD.

Comments:

My major concern for this paper is its unconvincing motivation and experiment results, especially when the approach is designed for deep learning.

The motivation for the proposed approach is not quite convincing. The authors said that “due to the large stochasticity, SGD with Nesterov’s momentum is not robust...This increased uncertainty slows down its convergence especially in the early stage of training and makes its single run less trustworthy” For image classification, Nesterov momentum is very popular and the final convergence values of different trails seems to be similar. It would be more convincing if the authors can provide practical evidence for supporting this claim.

It was claimed that Amortized Nesterov’s Momentum has “higher efficiency and faster convergence in the early stage without losing the good generalization performance”. What is the benefit or advantage for having faster early convergence without improving the final generalization performance?

The authors claim that “M-SGD2 is more robust and achieves slightly better performance”, in Figure 1a, however, it is really hard to tell the difference between M-SGD2 and M-SGD from Figure 1a.

The efficiency improvement in page 4 is really hard to follow for comparison with Algorithm 1 in page 3. Though m > 2 could reduce the number of operations in step 5 and 6, I don’t think this is a computational bottleneck. I believe these updates should be very fast in comparison with forward and gradient calculation. Making the 1% computation 50% faster does not mean more efficient. I would like to know how much computation cost can be saved with this modification. On the other hand, adding one more auxiliary buffer (scaled momentum) could significantly impact the training as the memory is often the limit.


In section 3.1, what is “identical iteration”? It is hard to compare AM2-SGD with AM1-SGD. It would be easier to follow if the side-by-side algorithm comparison can be shown early.

The section 4’s theoretical analysis based on the convex composite problem is not quite convincing. I am not sure how these results are related with the deep learning applications.

In the experiment section, the comparison of AM1/2-SGD with other baselines seems not quite consistent. The authors first state that they use all 0.1 learning rate and 0.9 momentum for all methods, however, the setting for M-SGD is using 0.99 momentum and different learning rate schedule. This makes the comparison not very meaningful, while AM1-SGD and AM2-SGD do not use learning rate restart. With so many differences, the advantage of AM1-SGD and AM2-SGD are not that different with M-SGD.  In the task of ImageNet-152, M-SGD even is better than AM1-SGD. This makes the conclusion that “AM1-SGD has a lightweight design, which can serve as an excellent replacement for M-SGD in large-scale deep learning tasks” not quite valid.

Minor: The author may assume readers maybe familiar Katyusha momentum, I think there may need more background about it.


**Experience Assessment:**

I have read many papers in this area.

**Review Assessment: Checking Correctness Of Derivations And Theory:**

I did not assess the derivations or theory.

**Review Assessment: Checking Correctness Of Experiments:**

I carefully checked the experiments.

**Review Assessment: Thoroughness In Paper Reading:**

I read the paper at least twice and used my best judgement in assessing the paper.

---

> ### Author Response · Authors · 2019-11-11
> **Response to Reviewer 2 (part 2/2)**
>
> [Experiment section]
> We have revised Section 5 to clarify that: For all the experiments in the paper, we aligned $(\eta, \beta)$ for AM1/2-SGD and M-SGD and thus they used the same learning rate schedulers. Our goal is to show their potentials of serving as alternatives to M-SGD. The restart (which is simply setting the buffers to the current iterate) is not performed on the LSTM experiment since the decay factor is small (i.e., 2) and the “decay on plateau” scheduler changes the learning rate frequently when converging. Based on the observations in Appendix A.4, the restart makes a difference only when the hyper-parameter setting is too aggressive.
>
> [Minor]
> We have shortened the “Connections with Katyusha” part and moved the details to the appendices.
>
> [1] Zeyuan Allen-Zhu. Katyusha: The first direct acceleration of stochastic gradient methods, JMLR 2018.
> [2] Ilya Sutskever, James Martens, George Dahl, Geoffrey Hinton. On the importance of initialization and momentum in deep learning, ICML 2013.

---

> ### Author Response · Authors · 2019-11-11
> **Response to Reviewer 2 (part 1/2)**
>
> Thank you for your valuable comments.
>
> We admit that the introduction in our original submission is not clear enough, which makes the proposed methods less appealing to practitioners. We have revised the introduction to emphasize the ‘easy-to-use’ feature from users’ perspective. Compared with M-SGD, AM1-SGD only has an additional integer $m$ to tune, and its learning rate $\eta$ and momentum parameter $\beta$ are strictly aligned with M-SGD. When $m=1$, AM1-SGD reduces to M-SGD. We conducted an extensive study to identify the benefits of setting $m$ besides $m=1$ and use the intuition from convex analysis to understand the difference.
>
> [Due to the large stochasticity, SGD with Nesterov’s momentum is not robust......]
> The purpose of this paragraph in the original submission is to point out the increased uncertainty of M-SGD in the first 60 epochs compared with SGD in Figure 2b. It is then shown in Figure 3 that AM1-SGD, while using an aggressive momentum (which achieves comparable final performance), is capable of reducing the uncertainty. We agree that the final convergence values of M-SGD are similar, which makes this improvement less appealing to practitioners that work on image classification, and thus we have removed the claim in the revision. However, there are deep learning problems such as reinforcement learning that are known to suffer from the robustness issue (or unstable issue in other works). In these areas, Adam is always the choice of optimizer and many models are tuned for it. We did not derive an adaptive variant for AM1-SGD since to fully explore the amortization technique already requires extensive study. Our work provides a new trick to ease the robustness issues and can serve as a foundation work for future development.
>
> [Faster early convergence without improving the final generalization]
> Since the underlying acceleration mechanism is still in Nesterov’s style (in convex analysis, Nesterov’s momentum is at the heart of all provable accelerated first-order methods [1]), we did not expect AM1-SGD to outperform M-SGD especially in our set-ups where $\eta$ and $\beta$ are strictly aligned. We observed improved early convergence, which is a benefit that comes from simply setting $m$ slightly differently from $m=1$.
>
> [M-SGD2 and M-SGD]
> Thanks for pointing out this issue. We have fixed it by adding detailed data in "M-SGD vs. M-SGD2" in the revision.
>
> [Efficiency improvement]
> We have moved Algorithm 1 to the page of "Efficiency" to improve readability.
> We recorded all the wall-clock time in the CIFAR experiments. However, we observed that even on the same type of GPUs, the running times fluctuate a lot and do not exhibit a clear trend with increasing $m$. Thus we did not report them in the paper. The running time is improved by about 2%-3% for AM1-SGD ($m=5$) compared with M-SGD (measured on the same GPU and using the same random batches). While not being the computational bottleneck, the improved efficiency is another benefit by simply setting $m$ slightly differently from $m=1$.
>
>
> [Adding one more auxiliary buffer]
> One more buffer is a drawback but we think that in most cases, it is acceptable since Adam also uses 2 buffers.
>
> [Identical iteration]
> “Identical iteration” means that the workload varies for different iteration $k$. For AM1-SGD, it is due to the if-clause at Step 4.
> Algorithms 1 and 2 are presented in ways that are easier to implement. We will try to make the connection between AM1-SGD and AM2-SGD clearer in the next revision.
>
> [Section 4’s theoretical analysis]
> We have revised the first paragraph in Section 4 to emphasize that the purposes are to understand the difference between AM1/2-SGD and M-SGD (or the effects of setting $m$ besides $m=1$) and to provide some intuition on tuning $m$. Nesterov’s momentum was migrated from convex optimization to deep learning by Sutskever et al. [2] and it turned out to be quite successful. We actually followed Sutskever et al. [2] to design tricks in convex analysis and explore their benefits in deep learning applications.

---

### Official Review · AnonReviewer4 · 2019-11-03
**Official Blind Review #4**

**Rating:** 1

**Review:**



This paper proposes two variants of Nesterov momentum that maintains a buffer of recent updates. The paper proves optimal convergence in the convex setting and makes nice connections to mirror descent and Katyusha.

I vote to reject the submission, with my main concerns being with the experimental results. I would consider raising my score if my concerns are addressed.

Concerns
-The learning rate schedule on the CIFAR experiments is unconventional. The original ResNet paper trains for 64k iterations (roughly 160 epochs). It’s standard to train for at least 200 epochs (see schedule from Smith et al.). Do the results hold under the standard schedule with careful tuning for the baselines?
-The discussion on “Train-batch loss vs. Full-batch loss” in Section 2 is unclear. On smaller datasets, it is feasible to perform evaluation at the end of the epoch on the entire batch. Furthermore, reporting test accuracy couples optimization and generalization, and I am not sure what is meant by the statement ``test accuracy is too informative.”
-Reporting the peak test accuracy is strange. In general, we do not have access to the test set. It’s more natural to report the final test accuracy or have a holdout set to determine an iteration for evaluation.
-It’s not clear how significant the results on ImageNet and PTB are. Namely, a comparison to AggMo and/or QHM would be good, since the flavor of these algorithms is quite similar. Experiments in the AggMo paper suggest that AggMo performs better on PTB. In the comparison given in Appendix A3, it seems like AggMo performs slightly better throughout training. SGD should also attain better performance with learning rate tuning on ImageNet.
-I’m not sure how useful “Test Accuracy STD%” is useful as a metric since it is influenced heavily by the learning rate and its schedule. Tail averaging schemes in general seem like they would increase “robustness.” In addition, there seem to be situations where a higher final variance is beneficial (just run the method for longer and you can find a better solution). It would be nice to expand the discussion on the notion of robustness defined in the paper.

Minor Comments
-The dashed line in figure 1b) is hard to read. I would recommend removing the grid lines and make the colors more differentiable.
-Algorithm 1: use of both assignment and equality operator? Whereas other boxes use equality
-Spacing looks a bit off in parts of the paper. 1) after the first sentence in the introduction 2) “generic optimization layer (Defazio, 2018) . However”)
-M-SGD and M-SGD2 can be potentially confusing and are not too informative as acronyms.
-Remark on Theorem 1b: depicts does not seem like the right word

Smith, S. L., Kindermans, P. J., Ying, C., & Le, Q. V. (2017). Don't decay the learning rate, increase the batch size. arXiv preprint arXiv:1711.00489.

**Experience Assessment:**

I have published one or two papers in this area.

**Review Assessment: Checking Correctness Of Derivations And Theory:**

I assessed the sensibility of the derivations and theory.

**Review Assessment: Checking Correctness Of Experiments:**

I carefully checked the experiments.

**Review Assessment: Thoroughness In Paper Reading:**

I read the paper at least twice and used my best judgement in assessing the paper.

---

> ### Author Response · Authors · 2019-11-09
> **Response to Reviewer 4 (part 2/2)**
>
> [Robustness]
> We think that to evaluate and to understand the behavior of an optimizer, it is important to measure the uncertainty in its convergence. The STD is indicating how aggressive a hyper-parameter setting is for an optimizer.
>
> An example is the comparison of SGD and M-SGD (Appendix C.1): when we align the effective learning rate of SGD and M-SGD, they produce similar performance (training ResNet34 on CIFAR-10). This phenomenon is used to question the effect of momentum in [1]. However, by measuring the STD, we see that the difference is that SGD has an average STD at 2.205% while M-SGD only has 1.040%, which indicates that this learning rate is too large for SGD or their settings are not at the same level of “aggressiveness”. This observation is suggesting that M-SGD still has room for more aggressive hyper-parameter settings, which can potentially increase the performance. In this sense, AM1-SGD has more room for “aggressiveness”, i.e., by grid-searching for larger $\eta$ or $\beta$ that are different from those of M-SGD. We didn’t do so since the current set-ups already achieve comparable final performance as M-SGD and are easy to use. We will include more discussion on robustness in the next revision.
>
> Tail-averaging does improve robustness, which is intuitively the difference between Option II and Option I in Table 1. What we emphasized is that the amortized momentum (Option I, no tail-averaging) also increases robustness.
>
> [Higher final variance is beneficial]
> From the intuition of the theory parts, $m$ is trading acceleration for variance control. In a concrete situation, users can determine the amount of variance control they need. Perhaps an adaptive choice of $m$ can be better.
>
> [Minor Comments]
> Thanks for pointing out those issues. We have fixed them in the revision.
>
> [1] Jerry Ma and Denis Yarats. Quasi-hyperbolic momentum and adam for deep learning, 2019
> [2] Liangchen Luo, et al. Adaptive gradient methods with dynamic bound of learning rate, 2019

---

> > ### Comment · AnonReviewer4 · 2019-11-13
> > **Response to authors**
> >
> > Thank you for the detailed response and update to the draft. Some thoughts:
> >
> > I think it’s better practice to report final test accuracy, even if does not illustrate the tradeoff as clearly.
> >
> > [Robustness]
> > I think this is an interesting point that the settings are not at the same level of “aggressiveness.” Do you have empirical evidence to back up this claim? It would be great (and strong support for your paper) if you could show “more aggressive hyper-parameter settings” achieving higher final performance for your optimizer
> >
> > Lastly, one point I missed in the original review is that Figure 1 is using the same learning rates for SGD and M-SGD. While this may be justified by theory, SGD should perform better if you use the same effective learning rate as in Ma & Yarats? I’m also not exactly sure what the goal of Figure 1 and its discussion is.
> >
> > Thanks again for the response.

---

> > > ### Author Response · Authors · 2019-11-14
> > > **Response to Reviewer 4**
> > >
> > > Thank you for your comments.
> > >
> > > We have changed to report the final test accuracy in all the experiments in the latest revised paper.
> > >
> > > [Robustness]
> > > We did a relevant experiment in Appendix A.2, where a larger $\beta$ is used. It shows that the amortized momentum does allow for more aggressive settings (when $\beta=0.99$, M-SGD is much more unstable than AM1-SGD). However, it seems that on the task of training ResNet34 on CIFAR-10, more aggressive settings do not result in a higher final performance. Another evidence is that AggMo performs the best when $K=2$ on this task, which is a much less aggressive setting than its suggested $K=3$.
> > >
> > > We agree that it is a strong point if we can find a task where AM1-SGD performs better with an aggressive setting, which M-SGD cannot adopt. It shows the potential of AM1-SGD.
> > > However, the motivation of this paper is to design a favorable alternative to M-SGD and users know exactly what to expect when switching to it (by setting $m$ larger than $1$). This benefits users who do not have enough computational budget to do grid search.
> > >
> > > [Figure 1]
> > > We have revised the “How to compare SGD and M-SGD?” part to clarify that the usage of SGD in this paper is for reference. That is, we regard momentum as a technique that is built upon plain SGD. We seek to understand what has changed when applying Nesterov’s momentum or amortized momentum. We have changed “How to compare SGD and M-SGD?” to “SGD and M-SGD” to make it less like a performance comparison.
> > >
> > > Figure 1a shows that adding Nesterov’s momentum hurts the performance in the first 60 epochs but accelerates the final performance. By comparing to Figure 2b, we see that the amortization technique eases this issue without losing the final performance. Figure 2b shows that Nesterov’s momentum slightly increases the uncertainty while Figure 3 & Table 1 show that the amortized momentum reduces the uncertainty. We have clarified these two points in the revision. Figure 1c is used to point out that train-batch loss is too inaccurate to be an evaluation metric. Moreover, the difference between OM-SGD and M-SGD is similar to the difference between Option II and Option I (since the options are designed in a similar manner).

---

> ### Author Response · Authors · 2019-11-09
> **Response to Reviewer 4 (part 1/2)**
>
> Thank you for carefully reading our paper and giving valuable feedback. We address your concerns as follows.
>
> [About the CIFAR-10 experiments]
> The choices of the 90 epochs training and the learning rate scheduler (decay 10-fold every 30 epochs) followed the parameter sweep experiments in QHM [1]. We swept the choices of $m$ and Options in Figure 2a (the complete sweeping results are given in Table 4). In order to study the robustness, for each choice of method and $m$, we ran 5 seeds, which means that totally 450 epochs were trained for an entry of Table 4. Then, to provide a stronger justification, we fixed $m=5$ and ran 20 seeds, which means that totally 1800 epochs were trained for each curve in Figure 2b.
>
> Since the 90 epochs training is not conventional, we chose the models, ResNet34 and PreActResNet18 (in Figure 4 / 12, which is also the choice in QHM), that can achieve quite decent performances in 90 epochs. For example, the ResNet34 experiments in [2] were trained for 200 epochs and the final performance result was roughly the same as ours.
>
> [Train-batch loss vs. Full-batch loss]
> We have revised this part to improve clarity. We did a full-batch loss experiment for training PreActResNet18 on CIFAR-10 in Figure 4. Comparing the full-batch loss curves and test accuracy curves (i.e., Figure 1c vs. Figure 1a, Figure 4 vs. Figure 12), the convergence (and robustness) on test accuracy is very similar to the convergence on full-batch loss (like being flipped and scaled).
>
> [Reporting peak test accuracy]
> We checked the final test accuracy: In the 20 runs experiments (Figure 2b), the peaks are all reached at the last epoch. In the 5 runs sweeping results (Figure 2a), AM1-SGD looks better in terms of final test accuracy. However, the trade-off between final test accuracy and $m$ is not as clear as peak test accuracy with $m$. In the ImageNet experiments (Figure 5b), the results are similar in terms of final test accuracy. We can report all the final test accuracies if necessary.
>
> [The ImageNet and PTB experiments]
> We clarified the goal of these experiments in the revision, which is to show the potentials of AM1/2-SGD of serving as alternatives to M-SGD.
>
> It is hard to compare AM1-SGD with QHM and AggMo without conducting extensive parameter sweeping experiments. We list some differences here:
> -  QHM
>  The fairness in all the experiments of QHM is that all the methods used the same effective learning rate. However, it is not clear what is the effective learning rate for AM1/2-SGD. We assumed that the amortization techniques do not change the effective learning rate of M-SGD and conducted the experiments in Appendix A.3.
> The hyper-parameter setting of QHM involves a scaled learning rate, which is not common, and two floats, which are not easy to tune without any insight on the effect. In comparison, the hyper-parameter setting of AM1-SGD is built upon M-SGD, which has extensive existing results to refer to, and for the choice of $m$, we provide intuition based on convex analysis, i.e., the effect of $m$ is trading acceleration for variance control.
> -  AggMo
> AggMo maintains $K$ velocity vectors and combines them in each iteration, which means that its memory cost (similar to AM2-SGD) and iteration cost scale with $K$. AggMo has totally $K+1$ tunable hyper-parameters and their paper suggests using an exponential setting with a fixed scale $a=0.1$ and varying only $K$. From our experience, when conducting the results in Appendix A.3, the performance of AggMo is highly sensitive to the choice of $K$. In comparison, the performance of AM1-SGD is stable for a wide range of $m$ (Figure 2a, 6) and we prove that it converges for any valid choice of $m$ in the convex setting.
>
> SGD in this paper is mainly involved as a reference. For example, in “A momentum that increases robustness” in Section 3, we use the robustness of SGD to support the claim that using amortized momentum (AM1-SGD with Option I, no tail-averaging) increases robustness.

---

### Author Response · Authors · 2019-11-08
**Revision on clarifying the motivation in the introduction**

Thank you all for your detailed reviews and valuable comments.

One major concern is on the motivation, which we admit that we should have made it clearer. We have revised our paper to clearly state that our motivation is to design a favorable alternative for M-SGD. We realized that our original paper emphasizes too much on the robustness, which undermines other advantages of AM1-SGD, especially ignoring the "easy-to-use" feature of AM1-SGD. In the revised version, we clarified the above in the introduction and listed all the pros and cons of AM1-SGD. We hope the changes would allow reviewers and readers to better appreciate the contributions of our work.

Other modifications according to the reviews are given as follows:
- Revised abstract and conclusion for consistency.
- Emphasized the understanding parts in "Convergence Results".
- Clarified the goal of ImageNet and LSTM experiments in the first paragraph of Section 5.
- Revised the discussion on “Train-batch loss vs. Full-batch loss”.
- Made Figure 1b) clearer.
- Added detailed data in "M-SGD vs. M-SGD2".
- Changed M-SGD2 to OM-SGD, where 'O' stands for "Original".
- Improved the presentation of Algorithm 1, and moved it to the page of "Efficiency".
- Added more descriptions in "A momentum that increases robustness".
- Fixed the spacing issues.
- Made "Options and convergence" for AM2-SGD clearer.
- Shortened Section 4 by moving the parts related to giving unified proofs to Appendix B.2.
- Added more explanations to the assumptions.
- Changed all $O(1/L)$ to explicit upper bounds.
- Cited missing references.

The pros and cons of AM1-SGD are listed as follows:

PROS:
- Easy-to-use. AM1-SGD uses exactly the same values for $(\eta, \beta)$ as M-SGD. It has only one additional integer $m$ to choose and by setting $m=1$, it recovers M-SGD. (In all the experiments in the paper, $(\eta, \beta)$ for AM1/2-SGD was aligned with M-SGD)
- Increasing $m$ improves robustness. This is an interesting property: what AM1-SGD (Option I, no tail-averaging) does is to inject a very large momentum into plain SGD every $m$ iterations. It turns out that this momentum not only provides acceleration, but also helps the algorithm become more robust than plain SGD.
- Increasing $m$ reduces (amortized) iteration complexity.
- A suitably chosen $m$ boosts the convergence rate in the early stage of training and produces comparable final generalization performance.
- It is easy to tune $m$. The performance of AM1-SGD is stable for a wide range of $m$ (Figure 2a, 6) and we prove that it converges for any valid choice of $m$ in the convex setting.
- In the convex setting, AM1-SGD is proved optimal, just like M-SGD.
- AM1-SGD is more robust to large $\beta$ (Appendix A.2).

CONS:
- One more memory buffer required compared with M-SGD.
- The learning rate schedulers issue, which we addressed by performing restarts (i.e., to set the momentum buffers to the current iterate).

---

### Decision · Program_Chairs · 2019-12-19

**Decision:**

Reject

**Comment:**

This paper introduces a variant of Nesterov momentum which saves computation by only periodically recomputing certain quantities, and which is claimed to be more robust in the stochastic setting. The method seems easy to use, so there's probably no harm in trying it. However, the reviewers and I don't find the benefits persuasive. While there is theoretical analysis, its role is to show that the algorithm maintains the convergence properties while having other benefits. However, the computations saved by amortization seem like a small fraction of the total cost, and I'm having trouble seeing how the increased "robustness" is justified. (It's possible I missed something, but clarity of exposition is another area the paper could use some improvement in.) Overall, this submission seems promising, but probably needs to be cleaned up before publication at ICLR.